# Fast Training of Neural Lumigraph Representations using Meta Learning

**Alexander W. Bergman**
Stanford University
awb@stanford.edu

**Petr Kellnhofer**
Stanford University
pkellnho@stanford.edu

**Gordon Wetzstein**
Stanford University
gordon.wetzstein@stanford.edu

computationalimaging.org/publications/metanlr/

## Abstract

Novel view synthesis is a long-standing problem in machine learning and computer vision. Significant progress has recently been made in developing neural scene representations and rendering techniques that synthesize photorealistic images from arbitrary views. These representations, however, are extremely slow to train and often also slow to render. Inspired by neural variants of image-based rendering, we develop a new neural rendering approach with the goal of quickly learning a high-quality representation which can also be rendered in real-time. Our approach, MetaNLR++, accomplishes this by using a unique combination of a neural shape representation and 2D CNN-based image feature extraction, aggregation, and re-projection. To push representation convergence times down to minutes, we leverage meta learning to learn neural shape and image feature priors which accelerate training. The optimized shape and image features can then be extracted using traditional graphics techniques and rendered in real time. We show that MetaNLR++ achieves similar or better novel view synthesis results in a fraction of the time that competing methods require.

## 1 Introduction

Learning 3D scene representations from partial observations captured by a sparse set of 2D images is a fundamental problem in machine learning, computer vision, and computer graphics. Such a representation can be used to reason about the scene or to render novel views. Indeed, the latter application has recently received a lot of attention (e.g., [1]). For this problem setting, the key questions are: (1) How do we parameterize the scene, and (2) how do we infer the parameters from our observations efficiently? With our work, we offer new solutions to answer these questions.

Several classes of scene representation learning approaches have recently been proposed. One popular approach consists of coordinate-based neural networks combined with volume rendering, like NeRF [2]. Although these representations offer photorealistic quality for synthesized images, they are slow to train and render. Coordinate-based networks that implicitly model surfaces combined with sphere tracing–based rendering are another popular approach [3–6]. One benefit of an implict surface is that, once trained, it can be extracted and rendered in real time [5]. However, training these representations is equally slow as training volumetric representations. Finally, approaches that use a proxy geometry with on-surface feature aggregation are also fast to render [7, 8], but the quality and runtime of these methods is limited by the traditional 3D computer vision algorithms that pre-compute the proxy shape, such as structure from motion (SfM) or multiview stereo (MVS).

Here, we develop a new framework for neural scene representation and rendering with the goal of enabling both fast training and rendering times. To optimize the training time of our framework, we do not learn a representation network for the view-dependent radiance, as other neural volume

or surface methods do, but directly aggregate the features extracted from the source views on the surface of our learned proxy shape. To cut down on pre-processing times required by SfM and MVS, we optimize a coordinate-based network representing the proxy shape end-to-end with our CNN-based feature encoder and decoder, and learned aggregation function. A key contribution of our work is to combine this unique surface-based neural rendering framework with meta learning, which enables us to learn efficient initializations for all of the trainable parts of our framework and further minimize training time. Because our representation directly parameterizes an implicit surface, it can be extracted and rendered in real time. This representation thus incorporates both the fast training benefits from generalizing over shape representations and image features, and the fast rendering capabilities of implicit surface-based methods. We demonstrate training of high-quality neural scene representations in minutes or tens of minutes, rather than hours or days, which can then be rendered at real-time framerates.

## 2 Related Work

**Image-based rendering (IBR).** Classic IBR approaches synthesize novel views of a scene by blending the pixel values of a set of 2D images [9–18]. Recent IBR techniques leverage neural networks to learn the required blending weights [19–24]. These neural IBR methods either use proxy geometry, for example obtained by SfM or MVS [25, 26] or depth estimation [27], together with on-surface feature aggregation [7, 8], or use learned pixel aggregation functions [28, 29] for geometry-free image-based view synthesis. Our approach is closely related to the geometry-assisted and feature-interpolating view synthesis techniques. Many existing approaches, however, require the proxy geometry to be estimated as a pre-processing step, which can take minutes to hours for a single scene, preventing fast processing of novel views. Other approaches which use monocular depth estimation to re-project and aggregate features such as SynSin [27], are applied only to single input images, and cannot easily be scaled to multi-view data due to depth estimation inconsistencies between images of the scene. Instead, our approach estimates a coordinate-based neural shape representation from the input images. The shape representation is optimized end-to-end with the image feature extraction, aggregation, and decoding and is accelerated using meta learning.

**Neural scene representations and rendering.** Emerging neural rendering techniques [1] use explicit, implicit, or hybrid scene representations. Explicit representations, for example those using proxy geometry (see above), object-specific shape templates [30], multi-plane [2, 31–34] or multi-sphere [35, 36] images, or volumes [37, 38], are fast to evaluate but memory inefficient. Implicit representations, or coordinate-based networks, can be more memory efficient but are typically slow to evaluate [3–5, 39–55]. Hybrid representations make a trade-off between computational and memory efficiency [56–59]. Among these, NeRF [60] and related methods (e.g., [61–66]) use coordinate-based network representations and volume rendering, which requires many samples to be processed along a ray, each requiring a full forward pass through the network. Although recent work has proposed enhanced data structures [67–69], network factorizations [70], pruning [52], importance sampling [66], fast integration [71], and other strategies to accelerate the rendering speed, training times of all of these methods are extremely slow, on the order of hours or days for a single scene. DVR [3], IDR [4], NLR [5] and UNISURF [6] on the other hand, leverage implicitly defined surface representations, which are faster to render than volumes but are equally slow to train. While concurrent work has improved the applicability of these methods by addressing limitations of surface-based methods in general, such as removing the object mask requirement [6, 72, 73], these approaches are still slow to train as they rely on hybrid volumetric and surface formulations to bootstrap the neural surface training. Our approach builds on generalization approaches for neural scene representations to accelerate the training time of 2D-supervised neural surface representations, and thus can be applied alongside advanced training strategies for neural surfaces and make these representations even more applicable for practical view synthesis.

**Generalization with neural scene representations.** Being able to generalize across neural representations of different scenes is crucial for learning priors and for 3D generative-adversarial networks with coordinate-based network backbones [74–78]. A variety of different generalization approaches have been explored for neural scene representations, including conditioning by concatenation [39, 40], hypernetworks [47], modulation or feature-wise linear transforms [77, 79, 80], and meta learning [81, 82]. Inspired by these works, we propose a meta-learning strategy that

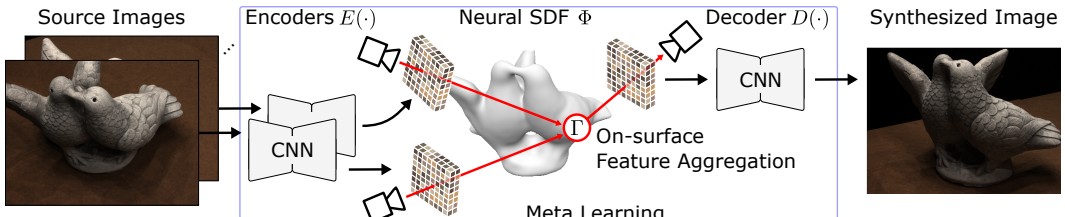

Figure 1: Overview of MetaNLR++.

allows us to quickly learn a high-quality neural shape representation for a given set of multi-view images. As opposed to the 3D point cloud supervision proposed by MetaSDF [81], we meta-learn signed distance functions (SDFs) using 2D images, and as opposed to meta-learned initializations for NeRF volumes [82], we operate with SDFs and features. Our approach is unique in enabling fast, high-quality shape representations to be learned from multi-view images that, once optimized, can be rendered in real time.

## 3 Method

### 3.1 Image formation model

In this section, we outline the NLR++ novel view synthesis image formation model, which is presented in Figure 1. NLR++ takes as inputs a set of images $\{I_i\}_{i=1}^{N}$ each with corresponding binary foreground object masks $\{M_i\}_{i=1}^{N}$ and known camera intrinsic parameters and extrinsic parameters collectively referred to as $C_i$. At output, NLR++ synthesizes an image $\hat{I}_t$ of the scene from the target viewpoint $C_t$.

Drawing inspiration from classic image-based rendering methods, we define the image formation model as a learned pixel-wise aggregation $\Gamma_\zeta(\cdot)$ of input image features $\{E_\xi(I_i)\}_{i=1}^{N}$ and their target viewing direction $V_t \in \mathbb{R}^{H \times W \times 3}$ on the surface of the object represented by the neural surface $\Phi_\theta$. To project the non-occluded, visible input features of $E_\xi(I_i)$ into the target view before aggregation, we define the function $\mathcal{P}_{i \rightarrow t}(E_\xi(I_i); C_i, C_t)$. These neurally aggregated features are then decoded into an image by decoder $D_\psi(\cdot)$:

$$\hat{I}_t = D_\psi(\Gamma_\zeta(\{\mathcal{P}_{i \rightarrow t}(E_\xi(I_i))\}_{i=1}^{N}, V_t)). \tag{1}$$

The feature encoder $E$ and decoder $D$ are implemented as resolution-preserving convolutional neural network (CNN) architectures [83, 84] with learned parameters $\xi, \psi$:

$$E_\xi : \mathbb{R}^{H \times W \times 3} \rightarrow \mathbb{R}^{H \times W \times d}, \quad D_\psi : \mathbb{R}^{H \times W \times d} \rightarrow \mathbb{R}^{H \times W \times 3} \tag{2}$$

To aggregate the input image features from $E$ into a target feature map to be decoded by $D$, we use a learned feature aggregation (or blending) function $\Gamma_\zeta$, which operates on the surface of our shape representation $\Phi$. To define the surface of our shape, we choose to use a SIREN [51] which represents the signed-distance function (SDF) in 3D space. This encodes the surface of the object as the zero-level set, $L_0$, of the network:

$$L_0(\Phi_\theta) = \{x \in \mathbb{R}^3 | \Phi_\theta(x) = 0\}, \quad \Phi_\theta : \mathbb{R}^3 \rightarrow \mathbb{R}. \tag{3}$$

The aggregation is performed on surface for each pixel of the target image $\hat{I}_t$ with camera $C_t$. To find the point in $L_0(\Phi_\theta)$ corresponding to each pixel ray, we perform sphere tracing on the neural SDF model $\mathcal{R}(\Phi_\theta, C_t)$, retaining gradients for the final step of evaluation [4, 5, 53, 54]. These individual rendered surface points are projected into the image plane of each of the $N$ input image views and used to sample interpolated features from the source feature maps for each pixel, which can be arranged into $N$ re-sampled feature maps corresponding to each input image $\{F_i\}_{i=1}^{N} \in \mathbb{R}^{N \times H \times W \times d}$. To check whether or not a feature is occluded, we use sphere tracing for each pixel from the input views $\{\mathcal{R}(\Phi_\theta, C_i)\}_{i=1}^{N}$, and ensure that the target view surface position projected into each of these surfaces is at the same depth as the source view surface position. Occluded features are then discarded. These three steps of sphere tracing, feature sampling, and occlusion checking make up the function $\mathcal{P}_{i \rightarrow t}(E_\xi(I_i))$ which outputs each of the re-sampled feature maps $\{F_i\}_{i=1}^{N}$.

Once the re-sampled feature maps $\{F_i\}_{i=1}^N$ have been generated, the aggregation function $\Gamma_\zeta$ aggregates them into a single target feature $F_t \in \mathbb{R}^{H \times W \times d}$ which can be processed by the decoder into an image. This is done by performing a weighted averaging operation on the input features using the relative feature weights $L \in \mathbb{R}^{N \times H \times W}$:

$$\Gamma_\zeta : \mathbb{R}^{N \times H \times W \times d} \times \mathbb{R}^{H \times W \times 3} \to \mathbb{R}^{H \times W \times d},$$

$$\{F_i\}_{i=1}^N, V_t \mapsto \Gamma_\zeta(\{F_i\}_{i=1}^N, V_t) = \sum_{i=1}^N (L_i / \sum_{j=1}^N L_j) \circ F_i = F_t, \qquad (4)$$

where $\circ$ is the Hadamard product between the feature and weight maps, broadcasted over the feature dimension $d$. The weight map $L$ used in the feature aggregation function $\Gamma_\zeta$ is obtained from an MLP $\gamma_\zeta$ which is applied pixel-wise to each of the $N$ re-sampled feature maps and each pixels target viewing direction:

$$[\{F_i\}_{i=1}^N, V_t] \mapsto \gamma_\zeta([\{F_i\}_{i=1}^N, V_t]) = L. \qquad (5)$$

Here, the dependence upon viewing direction allows for the modeling of view-dependent image properties. These pixel-wise operations making up $\Gamma$ result in a $H \times W \times d$ feature map which can be input into the decoder $D$.

The usage of features instead of pixel values directly allows $D$ some opportunity to inpaint and correct artifacts from imperfect geometry to create a photorealistic novel view, unlike methods which render pixels individually [4, 5]. Additionally, the use of the CNN encoder and decoder increases the receptive field of the image loss applied, allowing for more meaningful gradients to be propagated back into $\Phi_\theta$.

## 3.2 Supervision and training

Since NLR++ is end-to-end differentiable, we can optimize the parameters $\xi, \psi, \theta, \zeta$ end-to-end to reconstruct target views. For each iteration of training we sample a set of $k \leq N$ images $\{I_n\}_{n=1}^k$, and designate one of these images to be the ground-truth target image used for supervision $I_t$, and sample its corresponding binary object mask $M_t$ and parameters $C_t$.

Using the NLR++ image formation model, we generate a synthesized target image $\hat{I}_t$ from viewpoint $C_t$. The loss evaluated on the synthesized image consists of three terms:

$$\mathcal{L}(\{\hat{I}_t, \hat{M}_t\}, \{I_t, M_t\}, \Phi_\theta) = \mathcal{L}_R(\hat{I}_t, \{I_t, M_t\}) + \lambda_1 \mathcal{L}_M(\Phi_\theta, M_t) + \lambda_2 \mathcal{L}_E(\Phi_\theta). \qquad (6)$$

The image reconstruction loss $\mathcal{L}_R$ is computed as a masked L1 loss on rendered images:

$$\mathcal{L}_R(\hat{I}_t, \{I_t, M_t\}) = \frac{1}{\sum_p M_t[p]} \sum_{p|M_t[p]=1} |I_t[p] - \hat{I}_t[p]|. \qquad (7)$$

To quickly bootstrap the neural shape learning from the object masks, we apply a soft mask loss on the rendered image masks [4, 5]:

$$\mathcal{L}_M(\Phi_\theta, M_t) = \frac{1}{\alpha \sum_p M_t[p]} \sum_{p|M_t[p]=0 \vee \Phi_{min}[p]<\tau} \text{BCE}(\text{sigmoid}(-\alpha \Phi_{min}[p], M_t[p])), \qquad (8)$$

where the notation $\Phi_{min}[p]$ denotes the minimum value of the SDF $\Phi_\theta$ along the ray traced from pixel $p$, $\tau$ is a threshold for whether the zero level set $L_0(\Phi_\theta)$ has been intersected, and $\alpha$ is a softness hyperparameter. Finally, we regularize the shape representation to model a valid SDF by enforcing the eikonal constraint on randomly sampled points $p_i \in \mathbb{R}^3$ in a unit cube containing the scene:

$$\mathcal{L}_E(\Phi_\theta) = \frac{1}{P} \sum_{i=0}^P ||(||\nabla_p \Phi_\theta(p_i)||_2 - 1)||_2^2 \qquad (9)$$

However, to make our training more efficient, we augment the loss supervision schedule and batching strategy for our model. Specifically, for each sampled batch of $k$ images, instead of computing gradients for a single selected target image, we treat $l < k$ images as target images, and each of them from the other images in the batch. The number of target images $l$ is limited by the GPU memory

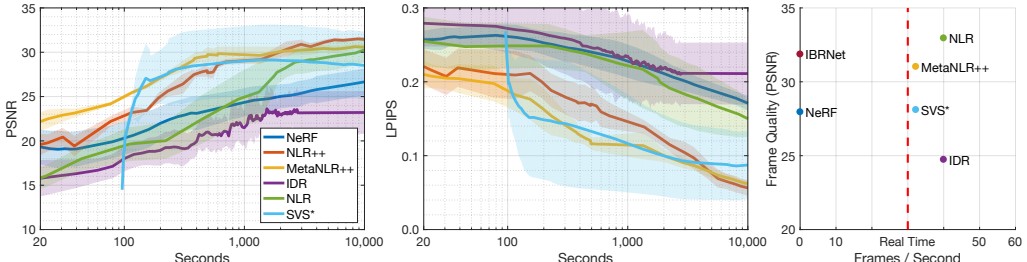

Figure 2: We demonstrate that at all training-times, MetaNLR++ is comparable to or outperforms all competitive representation learning methods, including both neural volumetric and surface representations in PSNR↑ (left) and LPIPS↓ [86] (center). We also plot the render time versus converged image quality, showing that MetaNLR++ can generate high-quality frames at real-time rates (right). The shaded area around each line represents the standard deviation of the method across three DTU scenes.

when computing the loss for each target image. Since all views must be sphere traced and passed through $E$ for a single target view, this additional batching only adds additional forward passes to $\Gamma$ and $D$, which are fast to evaluate. This batching strategy gives more accurate gradients for our model at each iteration. Additionally, since $\mathcal{L}_M$ requires us to find a minimum of $\Phi_\theta$ along a ray, it requires dense sampling of this network and accounts for most of the compute time of each forward pass. Thus, while optimizing NLR++, we propose to only enforce shape-related losses $\mathcal{L}_M$ and $\mathcal{L}_E$ for the first $t_1$ iterations, and then every $t_2$ iterations thereafter. This allows NLR++ to learn a shape approximation in the first $t_1$ iterations, and then further refine it as the feature encoding scheme with $E, D, \Gamma$ get significantly better. This is only possible since, unlike prior Neural Lumigraph work [4, 5], the appearance modeling is outsourced to the feature extraction from input images and is more independent from the current shape than a dense appearance representation in 3D space.

### 3.3 Generalization using meta learning

As our goal is to learn scene representations quickly, we use meta learning to learn a prior over feature encoding, decoding, aggregation, and shape representation using datasets of multi-view images. This prior is realized via the initialization of the networks $E_\xi, D_\psi, \Gamma_\zeta$ and $\Phi_\theta$ which dictates the network convergence properties during gradient descent optimization. For simplicity of notation and since we are meta-learning the initializations for all networks in NLR++, we define all NLR++ parameters as $\Theta = [\xi, \psi, \zeta, \theta]$.

Let $\Theta_0$ denote the NLR++ parameters at initialization, and $\Theta_i$ denote the parameters after $i$ iterations of optimization. For a fixed number of steps $m$ of optimization, $\Theta_m$ will depend significantly on the initialization $\Theta_0$, resulting in possibly significantly different NLR++ losses. We adopt the notation from [82], and will emphasize the dependence of parameters on initialization by writing $\Theta_m(\Theta_0, T)$, where $T$ is the particular scene which we would like to represent. We aim to optimize the initial weights $\Theta_0$ that will result in the lowest possible expected loss after $m$ iterations when optimizing NLR++ for an unseen object $T$, sampled from a distribution of objects $\mathcal{T}$. This expectation over objects is denoted as $E_{T \sim \mathcal{T}}$, resulting in the meta learning objective of:

$$\Theta_0^* = \arg\min_{\Theta_0} E_{T \sim \mathcal{T}}[\mathcal{L}(\Theta_m(\Theta_0, T))]. \tag{10}$$

To learn this initialization for $\Theta_0$, we use the Reptile [85] algorithm, which computes the weight values $\Theta_m(\Theta_0, T)$ for a fixed inner loop step size of $m$. Each optimization of NLR++ for $m$ steps for a different task sampled from $T_j \sim \mathcal{T}$ is referred to as an outer loop, and is indexed by $j$. To avoid having to compute second-order gradients through the NLR++ model, Reptile updates the initialization $\Theta_0$ in the direction of the optimized task weights with the following equation:

$$\Theta_0^{j+1} = \Theta_0^j + \beta(\Theta_m(\Theta_0^j, T_j) - \Theta_0^j), \tag{11}$$

where $\beta$ is the meta-learning rate hyperparameter. We label NLR++ with the meta-learning initializations applied MetaNLR++.

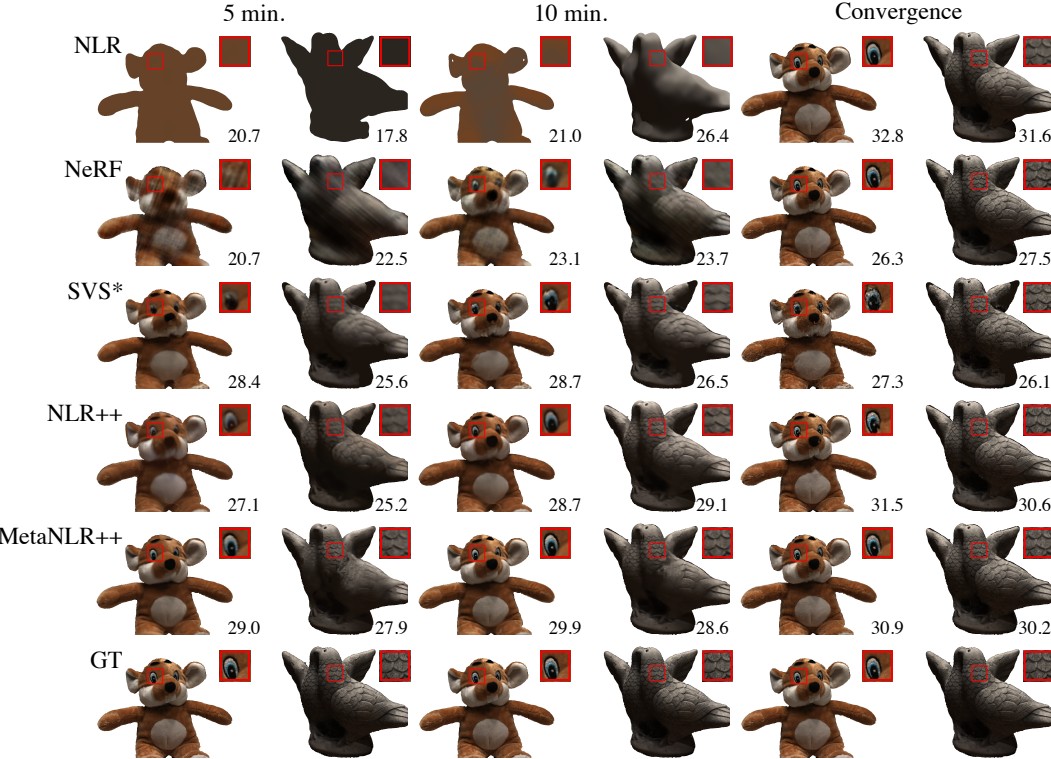

Figure 3: Novel views synthesized using various methods after a set training time. MetaNLR++ outperforms other surface and volume representation methods, especially for a training time budget on the order of minutes, and does not sacrifice quality of the final converged result.

### 3.4 Implementation details

The source code and pre-trained models are available on our project website, and the full set of implementation details including hyperparameters, training schedules, and architectures are described in our supplement for each of the various datasets evaluated on. We implement MetaNLR++ in PyTorch and use the Adam [87] optimizer for all optimization steps of NLR++, including for the inner-loop in meta learning, with a starting learning rate of $1 \times 10^{-4}$ for $\Phi$ and $5 \times 10^{-4}$ for $E, D, \Gamma$. We use $\alpha = 50$, $\tau = 1 \times 10^{-3}$, and $\beta = 1 \times 10^{-1}$ as starting hyperparameter values, which are progressively decayed (or increased in the case of $\alpha$) through training (full schedules are described in the supplement). We use shape loss training hyperparameter values of $t_1 = 50$ and $t_2 = 7$, and loss weight parameters of $\lambda_1 = 1 \times 10^2/\alpha$, $\lambda_2 = 3$. In the case of NLR++, we initialize $\Phi$ as a unit sphere of radius $0.5$ by pre-training our SIREN to represent this shape. We train each of our models on a single Nvidia Quadro RTX8000 GPU. We also use an Nvidia Quadro RTX6000 GPU for rendering and training iteration time computation. In total, we have an internal server system with four Nvidia Quadro RTX8000 GPUs and six Nvidia Quadro RTX6000 GPUs, of which we used a subset of three RTX8000s and one RTX6000.

## 4 Experiments

**Baselines.** Our main contribution is the rapid learning of a representation which can be used to render high-quality novel views of a scene in real time. We demonstrate this by comparison to several state-of-the-art methods. Specifically, we evaluate the volumetric representation of NeRF [60], a mesh-based representation similar to SVS [8], the neural signed distance function-based representations of IDR [4] and NLR [5], and the image-based rendering of IBRNet [29]. For SVS [8] we use our own simplified implementation and denote it SVS*. Our implementation trains the same $E, D, \Gamma$ as in MetaNLR++ but we replace the learnable shape by a surface reconstruction from COLMAP [25, 26].

**Training time vs. quality trade-off.** For the following comparisons, we use the DTU dataset [4, 88], which has been made public by its creators, and contains multi-view images of various inanimate objects, none of which are offensive or personally identifiable. In Figure 2, we plot the average PSNR and LPIPS score of three held-out test views on three test DTU scenes as a function of training time as measured on a Nvidia Quadro RTX6000. Each of these representations are trained using only seven ground-truth views from the DTU scene. The meta-learned initializations $\Theta_0^*$ are optimized using complete view sets from another 15 DTU scenes, distinct from the testing scenes. In these plots, we see that MetaNLR++ maintains high reconstruction quality throughout the training process which results in predictable quality progression for time-constrained applications. Beyond PSNR we showcase results of the perceptual LPIPS metric [86] as we observe that PSNR is not robust to small inaccuracies in the object masks and prefers low-frequency images. This trade-off is exemplified in Table 1, where we show that MetaNLR++ is able to reach the 25dB and 30dB PSNR milestones faster than any other learned scene representation. The difference is particularly large for the volumetric method NeRF that aggregates many radiance samples along each rendered ray, and the implicit surface-based method NLR which must simultaneously optimize a neural surface and neural representation of color densely in 3D. In the case of NLR, this is because every time the shape representation is updated, the color representation must also update to correctly reflect the appearance on the modified surface of the shape, leading to a slow training time.

While IBRNet [29] is able to generate high-quality novel views very quickly, it cannot be turned into a pre-computed mesh-based or volume representation and requires input images for each rendered frame, and is more aptly considered a feed-forward image-blending method instead of a neural scene representation. SVS* is initially offset by the runtime of COLMAP. Afterwards, it quickly fits $E, D, \Gamma$, but its maximum performance is limited by the initial geometry. Con-

Table 1: Training time to reach specified PSNR level. The best times and PSNR values are bolded (second best is underlined) for methods which can render at real-time framerates. The training times are included for NeRF and IBRNet as well, which are incapable of fast rendering. IBRNet is pre-trained on multi-view data, and thus needs no further training to reach 25dB PSNR.

|  | 25dB PSNR | 30dB PSNR | Maximum PSNR |
|---|---|---|---|
| IDR | - | - | 24.73dB |
| NLR | 14.7 min. | 191.4 min. | **32.95dB** |
| MetaNLR | 2.1 min. | 176.8 min. | 31.71dB |
| SVS* | 2.1 min. | - | 28.19dB |
| NLR++ | 3.2 min. | 37.4 min. | 31.02dB |
| MetaNLR++ | **1.9 min.** | **22.5 min.** | 30.57dB |
| NeRF | 33.3 min. | - | 27.95dB |
| IBRNet | 0 sec. | 23.1 sec. | 31.86dB |

sequently, $D$ must learn to inpaint the resulting holes which results in over-fitting to training views. This is notable in Figures 2, 3 and Table 1, where the quality of novel views quickly saturates or even degrades. Additionally, the meshing step time scales with the number of input views, and it can take up to 2 hours for scenes with dense view coverage as reported in [8]. MetaNLR is provided as a baseline which applies our meta learning method to the NLR formulation. This shows that simply applying meta learning to NLR does not lead to nearly as fast of training to high quality when compared to MetaNLR++, and therefore the improvements in scene parameterization in NLR++ and the generalization via meta learning are necessary components which jointly contribute to achieving the desired goal of both fast training and rendering. Qualitative comparisons are shown in the supplementary document. We emphasize that learning a representation from only seven views is a difficult task. NeRF in particular has difficulty avoiding over-fitting to training views in this scenario. We show qualitative results in Figure 3, which highlight that MetaNLR++ performs progressively higher quality view synthesis using the learned features and geometry as the training advances.

**Training to convergence.** In Table 1 and Figure 3, we show that MetaNLR++ does not sacrifice on final converged quality for the sake of speed. Given unconstrained learning time, MetaNLR++ is still able to produce images which are competitive with converged results of state-of-the-art representations. As noted in Table 1, IBRNet is able to also perform high quality novel view synthesis, and can render images with 29.20dB without fine-tuning. However, since the rendering time is based off of neural volume rendering, it cannot generate frames at nearly a real-time rate. Additional comparisons are provided in the supplementary document. A quantitative evaluation of image quality and runtime at rendering time is shown in Figure 2, where we see that our surface-based method is able to render in real time, unlike neural volume rendering methods.

**Real-time rendering.** Because MetaNLR++ extracts the appearance directly from the source images and transforms them to the novel view using a compact shape model, we can greatly accelerate the rendering of our trained representation by pre-computing these components. First, we use the marching cubes algorithm [89] to extract a mesh as an iso-surface of the SDF. Second, we store the features computed from the set of input images by our encoder $E$ as well as their associated depth maps. At render time, we simply need to aggregate these features based on our target viewing direction, and evaluate the decoder on the aggregated features for each frame. For the DTU images and network architecture, the decoder takes $8.7$ms, and the aggregation takes $22.3$ms, so we are able to render frames at real-time rates as shown in Figure 2.

**Meta-learning ablation.** We investigate the efficacy of meta learning in speeding up learning of individual components of our representation. Specifically, we ablate the contribution of meta learning on the shape representation parameters ($\theta_0$), and feature encoding network parameters ($\xi_0, \psi_0, \zeta_0$).

Additionally, we also compare MetaNLR++ to a variant with the encoder and decoder replaced by a direct extraction of image RGB pixel values and with and without a learned aggregation function. Here we clarify that replacing the encoder and decoder with pixel values is still a variant of NLR++, where we have simply replaced the

Table 2: Meta-learning ablation study. We report average PSNR of synthesized views after 10 minutes for each method.

|  | Meta init. applied to: | | |
|---|---|---|---|
|  | No net. | $\Phi$ only | All net. |
| RGB pixels & fixed $\Gamma$ | 27.85dB | 28.13dB | 28.13dB |
| RGB pixels & learned $\Gamma$ | 26.82dB | 26.90dB | 26.91dB |
| CNN $E/D$ & fixed $\Gamma$ | 28.62dB | 29.16dB | 29.88dB |
| CNN $E/D$ & learned $\Gamma$ | 27.32dB | 28.48dB | **29.91dB** |

features with pixel values directly, and does not model a dense, continuous color function as in NLR. The results of this ablation study are shown in Table 2.

We see that meta-learning of each component contributes to fast learning of a high-quality representation (last row). The learned $\Gamma$ improves the performance for the learned $E/D$ but it decreases the performance for the directly extracted RGB values. This implies that the classical unstructured lumigraph blending [14] works well for direct pixel values, but the additional flexibility of the learned $\Gamma$ can be advantageous with deep features. The learned $\Gamma$ performs well when meta-learned but it has difficulty to accurately learn the angular dependencies in only 10 minutes of training otherwise.

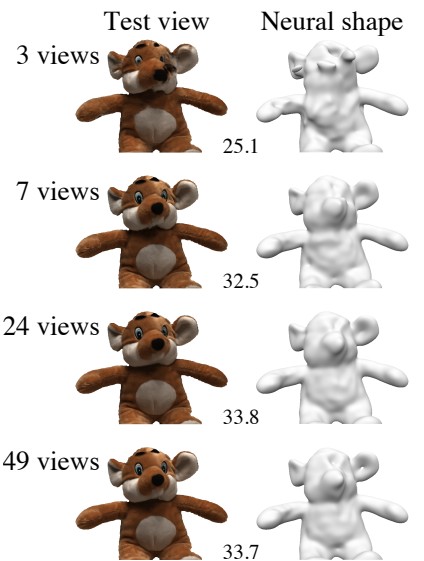

Test view    Neural shape

3 views    25.1

7 views    32.5

24 views    33.8

49 views    33.7

Figure 4: MetaNLR++ is robust to the number of views captured, which is essential in many applications where capturing a dense dataset is infeasible. In all cases, the learned $\Phi$ provides an adequate support for projection of our encoded appearance features.

More details of this experiment and additional results are available in the supplementary document.

**Input size ablation.** We illustrate the robustness of MetaNLR++ to low numbers of available training views in Figure 4. Here we see that our view synthesis quality decreases very gracefully with decreasing number of input views and it produces meaningful results even in the extreme case of three views. This is unlike COLMAP, which produces geometry with significant holes in occluded regions. These holes are responsible for the highly variable performance of SVS* shown in Figure 2. Additionally, while SVS* is able to quantitatively perform well when evaluated within the ground-truth mask, the holes in the geometry result in inaccurate rendering masks and thus severely limit novel view synthesis in practice. Additional results showing this phenomenon are included in the supplementary document. Since this ablation is performed on the DTU dataset, the input images capture only one side of the object. However, our method is applicable in scenarios where images are distributed $360°$ around the object. Additional results demonstrating this capability on the ShapeNet [90] dataset are included in the supplementary document. Our learned shape models are overly smooth relative to other methods and thus not particularly quantitatively accurate, but provide sufficient

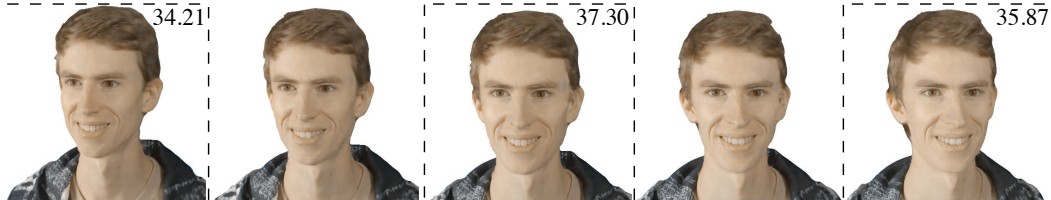

Figure 5: We compute additional results using the NLR dataset. The frames highlighted are supervised, and the intermediate frames are interpolated viewpoints for this MetaNLR++ model. Please see the supplement for additional results.

|  | 30 min. PSNR (LPIPS) | Convergence PSNR (LPIPS) |
|---|---|---|
| NLR++ | 28.22 (0.145) | 35.54 (0.046) |
| MetaNLR++ | **31.12 (0.083)** | **37.55 (0.034)** |

Table 3: Table comparing PSNR score on the supervised views of the NLR dataset. Since there are only 6 training views, we use all for training and report PSNR averaged on the three views shown in Figure 5, which shows that we are able to interpolate between these views.

accuracy for the image-based feature blending method to model appearance information observed in synthesized views. The smoothness of the learned shape is dependent on the capacity of the CNN architecture used for the feature encoder and decoder and coordinate-based network modeling the surface, as shown in the supplementary document. We opt to use higher capacity feature encoder and decoders in order to model sharp image details for novel view synthesis, which is responsible for the trade-off of quantitative accuracy of the neural shape representation.

**Additional results.**  To show that MetaNLR++ is robust to datasets beyond DTU, we evaluate using the multi-view dataset in NLR [5]. This dataset has been publicly released, and while the faces of the subjects in the dataset are personally identifiable, the subjects are the authors of NLR and have provided their consent of making this dataset public. The scenes in this dataset each have 22 multi-view images, taken with various cameras. We opt to use the final 6 views taken with high-resolution cameras to train our representation. We use 5 scenes in this dataset to learn the meta initialization $\Theta_0^*$, and test on a withheld sixth scene. The meta learning in this case leads to significantly improved performance in both PSNR and perceptual quality, despite only being able to learn a prior from a small number of scenes. This trend demonstrates that when the meta-training data more accurately covers the testing domain, the prior learned through meta learning is more effective at speeding up training time. This trend is further demonstrated in the supplementary document on the ShapeNet dataset, where the meta-training dataset is significantly larger and more closely related to the specialization data than in the case of DTU. With more comprehensive multi-view training datasets, this trend shows that MetaNLR++ could even further speed up neural representation training time.

In Table 3, we report our fit on the training frames after 30 minutes of training as benchmarked on our system. Additionally, we show interpolated frame results in Figure 5, demonstrating that MetaNLR++ is capable of generalizing to this scene and producing convincing novel-view synthesis results. Additional implementation information and results are included in the supplementary document.

## 5 Discussion and Conclusion

The fundamental problem of learning 3D scene representations from sparse sets of 2D images is rooted in machine learning, computer vision, and computer graphics. We provide an answer to this problem by proposing a novel parameterization of a 3D scene, and an efficient method for inferring these parameters from observations using meta learning. We demonstrate that our representation and training method are able to reduce representation training time consistently and render at real-time rates without sacrificing on image synthesis quality. This opens several exciting directions for future work in efficient training and rendering of representations, including using more advanced

generalization methods to learn representations in real time. With this work, we make important contributions to the field of neural rendering.

**Limitations and future work.**   While our method is able to produce compelling novel view synthesis results in a fraction of the time of other methods, we note that there are a few shortcomings. Specifically, in order to bootstrap the learning of a neural shape quickly, object masks are required to supervise the ray-mask loss. While these can be computed automatically for some data, this poses a challenge in cluttered scenes, or applications which could generalize to arbitrary scenes. Additionally, all of our experiments have used known camera poses to reconstruct the shape. Future work on jointly optimizing camera pose with our representation is certainly possible, and a step in the direction for general view synthesis. Our method is also limited by memory consumption, since the CNN feature encoder/decoders process the entire image at a time. This method could likely be improved by shifting to training and evaluating on image patches for higher resolution rendering. Finally, our method tends to produce overly-smoothed shape models, which, while beneficial for feature aggregation, are not always representative of high-frequency scene geometry. This highlights one fundamental trade-off: the capacity of the feature generation method versus the quality of the shape. With feature or color generation methods which are sufficiently regularized [4, 5], the model has no choice but to explain observed details in the neural shape model. We opt to utilize the full capacity of the CNN feature processing, as learning a detailed neural shape is slower than modeling fine details with features.

**Conclusion.**   The question of how to trade-off image synthesis quality with representation training and rendering time is of paramount importance to engineers, producers, or any other users of neural rendering technology. In the space of neural rendering methods, this work takes steps towards making representation learning and rendering more practical by optimizing this trade-off. Our novel scene parameterization and generalization method may provide a starting point for future work in optimizing this trade-off: speeding up representation training and rendering time and bringing modern neural rendering to the forefront of industry standard techniques.

# 6   Broader Impact

Methods such as MetaNLR++ for learning 3D scene representations from 2D representations allow for photorealistic image synthesis using only collections of other images. We have shown that MetaNLR++ improves upon the axes of training and rendering time of these representations, and as such may make it less computationally restrictive to use for individuals who want to learn and use 3D models from only collections of easily acquirable images. While this proliferation of neural rendering technology may be extremely helpful for many, it has the potential for misuse. As with any synthesis method, the technology could enable approaches to synthesis of deliberately misleading or offensive images, posing challenges similar to those posed by generative-adversarial models.

## Acknowledgments and Disclosure of Funding

Alexander W. Bergman was supported by a Stanford Graduate Fellowship. Gordon Wetzstein was supported by a PECASE by the ARO. Other funding was provided by the NSF (1839974).

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
