# Fast Training of Neural Lumigraph Representations using Meta-learning

## –Supplementary Document–

**Alexander W. Bergman**
Stanford University
awb@stanford.edu

**Petr Kellnhofer**
Stanford University
pkellnho@stanford.edu

**Gordon Wetzstein**
Stanford University
gordon.wetzstein@stanford.edu

## Contents

35th Conference on Neural Information Processing Systems (NeurIPS 2021).

# 1   Additional Implementation Details

As described in the main text, we plan to release all code used to obtain the results for our method. All data used has been made publicly available by their authors. We use PyTorch for all implementation, and evaluate all of our methods using our internal server consisting of four Nvidia Quadro RTX8000 GPUs and six Nvidia Quadro RTX6000 GPUs, which we used a subset of. Due to our limited resources and requirement of training our method and many baselines to convergence, we opt to report error bars with respect to multiple different testing scenes instead of different random seeds. Each of these evaluations is run with a randomly generated seed. Implementation details on network architectures and hyperparameters for the DTU [1, 2] and NLR [3] datasets are included in Sections 1.1 and 1.2 respectively.

## 1.1   DTU dataset

**Data.**   For each DTU scene, we use 7 of the ground truth 49 images for training. The view IDs of each of these images selected from the DTU dataset are: $[1, 9, 17, 25, 33, 41, 47]$. These views roughly image all parts of the object, but are not dense. The views held out for testing are views $[12, 32, 40]$. The images and ground truth masks are downsampled to resolution $800 \times 600$ as in [4], and all training and evaluation is performed on this resolution.

**Network architectures.**   To represent our neural shape $\Phi_\theta$, we use a 5-layer MLP with 128 hidden units per layer. Multiple architecture widths and depths were considered before finding this architecture which maximized the trade-off between evaluation speed and reconstruction quality for the DTU objects. The source image encoder $E_\xi$ is implemented as a ResNet [5], using the same architecture in SVS [4]. This network consists of a ResNet18 network with 4 residual blocks, each consisting of Conv2d-BatchNorm2d-ReLU-Conv2d-BatchNorm2d network layers, where the first Conv2d downsamples (or upsamples) the image resolution by half in each dimension. Each of the skip connection layers in the network consist of a single Conv2d-ReLU network layer pairing. The output number of features is set to $d = 16$. The target image decoder $D_\psi$ is implemented as a UNet [6], with 3 down/up-sampling layers. Each downsampling block consists of Conv2d-ReLU-Conv2d-ReLU-AvgPool2d network layers. The intermediate number of channels after each block is: $[64, 128, 256]$. The learned aggregation function is implemented as a 5-layer MLP with 32 hidden units per layer, which maps each feature in $\mathbb{R}^{16}$ and its target viewing direction in $\mathbb{R}^3$ to the aggregation weight in $\mathbb{R}$.

**Pre-training.**   The encoder and decoder networks are pre-trained using the FlyingChairs2 [7, 8] dataset. This dataset consists of pairs of images and ground-truth optical flow. To pre-train these networks, we apply two losses using image pairs and optical flow. The first loss feeds the image through the encoder, and the output features through the decoder, and ensures that the encoder and decoder are approximate inverses of each other. The second loss takes in the input image, warps the features, and then decodes the warped features into an image which is supervised by the warped image. This loss ensures that the encoder/decoder pair actually learn features representative of the image, and not to find some way to simply pass the input image through the feature bottleneck. This pre-training on the encoder and decoder especially helps NLR++ and SVS*, but MetaNLR++ is able to learn a prior over features using only meta learning on the DTU scenes. The shape network is pre-trained using a procedural sphere of radius 1.

**Training parameters.**   When optimizing a single MetaNLR++ or NLR++ model to represent a DTU scene, a learning rate of $\eta_1 = 1 \times 10^{-4}$ was used to train $\Phi_\theta$, and a learning rate of $\eta_2 = 5 \times 10^{-5}$ was used for $E_\xi, D_\psi, \Gamma_\zeta$. The increased learning rate for the encoding, aggregation, and decoding functions encourages the network to learn to model appearance with deep features rather than with geometry, which encourages faster convergence. Learning rate $\eta_1$ is decreased by half at iteration numbers: $[500, 1000, 3000, 7000, 15000, 31000]$. Learning rate $\eta_2$ is decreased by half every 2000 iterations, consistently.

Other training parameters relate to the loss function applied and determining if features are occluded. We use a $\mathcal{L}_M$ weight of $\lambda_1 = 1 \times 10^2/\alpha$, where $\alpha$ is the mask softness parameter. The value of $\alpha$ is set at 50, and is doubled at iteration numbers $[2000, 4000, 6000]$. This enforces the mask to be more and more binary as training goes on. We use a $\mathcal{L}_E$ weight of $\lambda_2 = 3.0$ for all DTU experiments, which does not decay. This is applied on randomly sampled points in the unit cube which our scene

representation $\Phi$ lies. To determine whether sphere-traced features are occluded, we check whether or not sphere-traced surface points from the target and source views have an L2-distance smaller than threshold $\tau$. We start with $\tau = 1 \times 10^{-3}$, which changes to $\tau = 1 \times 10^{-4}$ at 5000 iterations, and $\tau = 1 \times 10^{-5}$ at 10000 iterations. This encourages the occlusions to be more strict as the shape quality improves.

Additionally, as described in the main text, one key component which makes our method converge quickly is the ability to balance shape and feature network optimization. One way this is done is by not computing shape gradients on each iteration, and using previously cached surface points from each view to determine feature occlusions. For the first $t_1 = 50$ and every $t_2 = 7$ iterations thereafter, both shape and feature encoder/decoders are optimized as described in the image formation model. On these iterations, sphere-traced surface point locations are cached for each view. On iterations where the shape is not trained, these sphere-traced surface point locations are used to determine occlusions, and create a target image which can be used to update $E_\xi, D_\psi, \Gamma_\zeta$. All views are sphere-traced at initialization, in order to serve as the initial cached surface values. Although these points may not be completely accurate, there is benefit in optimizing the feature processing networks more often as the shape evolves, as they can better learn to blend features for this specific scene. When optimizing both shape $\Phi_\theta$ and feature processing networks $E_\xi, D_\psi, \Gamma_\zeta$, we use a batch size of all 7 input images. However, gradients for the shape are only computed for 4 of these images – the remaining 3 images use cached surface point locations, as previously described.

Other relevant implementation details are: occluded features are set to value zero, and thus contribute nothing to the weighted sum feature aggregation. The batch gradients, as previously described, are computed by summing loss terms for each of the 4 rendered target images. Finally, all optimization uses the Adam [9], and we have found that using other optimizers significantly decreases the performance. We expect that this is because we have not spent a significant amount of time tuning every hyperparameter (due to the large number of parameters), and the Adam optimizer is robust to some of these choices. We expect that with further hyperparameter tuning, our method could likely receive better results.

**Meta learning.**   We train the meta-initialization using 15 training DTU scenes, distinct from the scenes which we evaluate using. The initialization is learned using the Reptile algorithm [10], which simply updates the initialization in the direction of the optimized weights for $m = 64$ steps of fitting one of the training objects. This inner loop optimization is also done using the Adam optimizer, with the same training parameters as the previous section for the first 64 steps. However, $t_1$ is set to 64, which allows for shape optimization on every step of the meta-learning. The meta-learning rate is set to $\beta = 1 \times 10^{-1}$.

**Ablations.**   For the ablation study on the meta learning, we use the same parameters as normal training for each method, including for training the meta-learned initializations. When training the meta-learned initialization for only the shape, we use the same method as meta-learning all parameters, but only update the shape network weights.

For the ablation study on number of input views, we also use the same parameters as normal training for each method. For 3 views, we use views $[1, 25, 47]$. For 24 views, we use views $[1, 5, 9, 13, 17, 21, 25, 29, 33, 37, 41, 43, 47]$. For 49 views, we use all views $1 - 49$ but still withhold views $[12, 32, 40]$ (leading to only 46 training views). The PSNR is computed on withheld views $[12, 32, 40]$.

## 1.2   NLR dataset

**Data.**   For the NLR scenes, we use the last 6 ground truth images to train our representation. The view IDs of each of these images is: $[16, 17, 18, 19, 20, 21]$. This is done because these images are taken from the same camera, and using images from different cameras with a encoder and decoder which learn image priors may result in undesirable artifacts. These images all consist of human faces, an important application area for 3D representation learning. As in NLR, we do not withhold any specific views for testing, and instead qualitatively evaluate the interpolated view results. The images and ground truth masks are downsampled to resolution $800 \times 600$ as for the DTU dataset, and all training and evaluation is performed on this resolution.

**Network architectures.** We find that the default network architectures proposed for DTU lead to representations which build too much of the image appearance into the feature processing networks, and too little into the shape. This results in artifacts around the nose of subjects, where parts of the face should be occluded (see Figure 3). Thus, we propose to encourage the network to represent more of the high-frequency details using the geometry built into $\Phi$ instead of the feature processing $E, D, \Gamma$.

To represent our neural shape $\Phi_\theta$, we use a 5-layer MLP with 256 hidden units per layer. The source image encoder $E_\xi$ uses the same architecture as the DTU case. The target image decoder $D_\psi$ also uses a UNet to decode the features, but this UNet consists of 2 down/up-sampling layers with intermediate channel sizes after each block as $[32, 64]$. This smaller size prevents giving the decoder too much capacity to inpaint missing details, and overfit to the training images without refining the shape model. The learned aggregation function uses the same architecture as the DTU case.

**Pre-training.** Unlike in the DTU case, we do not pre-train the encoder and decoder networks using the FlyingChairs2 dataset. However, the shape network is still pre-trained using a procedural sphere of radius 1.

**Training parameters.** Most training parameters from the DTU case are re-used for the NLR case. One change regarding the batch size: since there are only 6 images in the dataset, the total image batch size is 6. However, the batch size of 4 for shape gradient computation remains the same. The other change is regarding the shape training versus iterations trade-off. Since these shapes require more fine detail to result in high-quality novel view synthesis, we opt to optimize the shape more often. Thus, the parameters $t_1, t_2$ are adjusted to $t_1 = 100, t_2 = 3$. This, along with the neural shape network size, affects the fastest possible speed with which we can represent these objects.

**Meta learning.** The meta-initialization is trained using 5 NLR scenes, distinct from the scene which we evaluate using. The initialization is learned using the Reptile algorithm with the exact same parameters as the DTU case.

### 1.3 ShapeNet dataset

**Data.** Each ShapeNet scene consists of 24 views at resolution $64 \times 64$. We withhold three views with IDs $[7, 16, 23]$ for testing, and use the remaining views for training. We randomly select 624 chair objects from the official ShapeNet training split to serve as meta-training data for the chairs split, and select 604 car objects from the official ShapeNet training split to serve as meta-training data for the cars split. We select 3 random objects from the official ShapeNet test set for the chairs and cars split respectively to serve as meta-testing samples.

**Network architectures.** To represent our neural shape $\Phi_\theta$, we use a 5-layer MLP with 128 hidden units per layer. This was selected to remain consistent with the experiments on the DTU dataset. Following this, the same architecture is used for the source image encoder $E_\xi$ and target image decoder $D_\psi$ and learned aggregation function as in the DTU experiments.

**Pre-training.** The ShapeNet experiments use the same pre-trained models as the DTU experiments - i.e. the encoder and decoder networks are pre-trained using the FlyingChairs2 dataset, and the shape network is pre-trained using a procedural sphere of radius 1.

**Training parameters.** Most training parameters from the DTU case are re-used for ShapeNet. Since the images are small and GPU memory is not a limitation, the batch size is increased to use all training 21 images when reconstructing each target image. At each iteration, the batch size is set to 10 for shape gradient computation (10 of the 21 images are treated as target images during each iteration). The parameters $t_1, t_2$ are adjusted to $t_1 = 2,000, t_2 = 5$. The large value of $t_1$ is acceptable due to the low resolution of the images, and thus the shape optimization through sphere tracing does not slow down each iteration of optimization by much.

**Meta learning.** The initialization is trained using the Reptile algorithm with the same parameters as the DTU case besides learning rate, which is decreased to $\beta = 2 \times 10^{-2}$. The meta-training and meta-testing splits are described in the section on data.

## 1.4 Timing

The timing for all methods was computed using an Nvidia RTX6000 GPU. For our method, we compute the timing by adding the time the forward pass takes, the time the loss computation takes, the time the backward pass takes, and the time that the optimization update takes. We compute distinct timing for iterations with and without shape optimization. For each of these types of iterations, we sampled 100 iteration times and averaged them to come to an iteration time. These values were then extrapolated to compute all timing results. The timing results for each of the methods at training time is:

> NeRF [11]: 0.40 sec/iteration.
>
> IBRNet [12]: 0.33 sec/iteration.
>
> NLR [3]: 2.21 sec/iteration.
>
> SVS* [4, 13, 14]: 1.38 sec/iteration after 96 second mesh computation time.
>
> IDR [1]: 0.2 sec/iteration.
>
> NLR++/MetaNLR++: 2.19 sec/iteration with no shape optimization, 7.69 sec/iteration with shape optimization.

Note that while MetaNLR++/NLR++ iterations appear to take significantly longer than those of other methods, these methods require significantly less iterations to converge. This is because, when compared to NLR or IDR, MetaNLR++ optimizes the representation for multiple entire images instead of randomly selected rays. The ray-batch size is significantly larger, resulting in longer iteration times.

The timing results for each of the methods at rendering time is computed assuming that the mesh and encoders can be pre-computed. Thus, this only requires learned aggregation and decoder evaluation, for our method and SVS*. For NeRF and IBRNet, the full forward pass must be ran on a full-resolution image. For IDR and NLR, this time is based on obtaining the pre-computed lumigraph, as described in NLR.

> NeRF: 32 sec/frame.
>
> IBRNet: 33.3 sec/frame.
>
> NLR: 0.025 sec/frame.
>
> SVS*: 0.031 sec/frame.
>
> IDR: 0.025 sec/frame.
>
> NLR++/MetaNLR++: 0.031 sec/frame.

## 1.5 Sphere tracing

We use the sphere tracing implementation published in [1]. This implementation uses bidirectional sphere tracing to find the intersection of the surface defined by $\Phi_\theta$ and a ray. We limit the sphere tracer to only 8 steps, and mark rays with SDF value within $5 \times 10^{-5}$ of 0 as converged. To find the minimum value along each ray necessary for $\mathcal{L}_M$, we densely sample 40 evenly spaced SDF values along a ray.

## 1.6 Result details

Here we describe additional noteworthy details of how the results in the main paper were computed.

**Figure 2.** The PSNR and LPIPS scores are computed on saved models throughout the training process after the convergence. These values smoothed values $\hat{v}_i$ were obtained using exponential moving average smoothing on the original measured values $v_i$, defined by:

$$\hat{v}_i = \begin{cases} v_i & i = 1 \\ \alpha v_i + (1 - \alpha)\hat{v}_{i-i} & i > 1, \end{cases} \tag{1}$$

where we use smoothing parameter $\alpha = 0.8$ for all methods besides IDR, which uses $\alpha = 0.3$ due to the much higher variance between plot points. Our definition of the smoothing parameter $\alpha$ implies that smaller $\alpha$ results in more smoothing, and larger $\alpha$ results in less dependence on previous values and thus less smoothing.

We train all methods until convergence. In the figure, IDR converges prior to the $10,000$ total seconds. Thus, we extrapolate these times using the average of the last 7 iteration results. This results in the straight line plot shown for the IDR bar in this figure.

**Table 1.** To find the time to reach specified dB PSNR for each model, we evaluate the PSNR for all saved models. We then use the lowest model iteration which reaches a specific metric, and compute the convergence time as a function of iterations. While it is computationally infeasible to evaluate all metrics for each model iteration, we save models as often as possible for each method. Here, we describe the model saving schedule, where notation $[n, m]$ means we have a model every $n$ iterations until total iteration $m$ is reached.

> NeRF: $[1, 10]$, $[10, 100]$, $[100, 1000]$, $[1000, 10000]$, $[10000, \text{end}]$
>
> IBRNet: $[1, 10]$, $[10, 100]$, $[100, 1000]$, $[1000, 10000]$, $[10000, \text{end}]$
>
> NLR: $[5, 100]$, $[50, 1000]$, $[200, \text{end}]$
>
> SVS*: $[5, 50]$, $[25, 250]$, $[250, \text{end}]$
>
> IDR: $[100, \text{end}]$
>
> NLR++/MetaNLR++: $[5, 50]$, $[25, 250]$, $[250, \text{end}]$

**PSNR/LPIPS Computations.** All PSNR computations are computed using the ground truth image masks and the method used in the code of SVS [4]. This method computes PSNR by element-wise multiplying (defined as $\circ$) ground truth and target image by the binary mask, and then computing PSNR on these images:

$$\text{PSNR}(\hat{I}_t, I_t, M_t) = 20 \log_{10}(1.0) - 10 \log_{10}((M_t \circ (\hat{I}_t - I_t))^2). \tag{2}$$

While this method biases the PSNR higher by using the masked values for error computation, it is consistently used for all baseline methods and thus comparisons are standardized. LPIPS metrics are also computed on the masked images which have also been multiplied by the ground truth mask, $\hat{I}_t \circ M_t$.

## 2 Baseline Implementation Details

**NeRF.** We use the NeRF implementation provided by the authors [11]. The training and evaluation were done with the same set of views and in the same $800 \times 600$ pixel resolution as for the scene fitting in our own method. Inspired by the provided *config_fern.txt* configuration, we evaluate 64 samples along each ray in the coarse phase and 128 samples in the fine phase and we process 2,048 rays in each batch. The training to convergence for each test scene was stopped after 200,000 solver steps as recommended by the authors.

**IBRNet.** We use the IBRNet implementation provided by the authors [12]. We use parameters from the provided *pretrain.txt* configuration to learn the generalized model for 250,000 solver steps on the same set of 15 training DTU scenes as for our own method. We then fine-tune the pre-trained model on the seven training views of a specific test scene for another 60,000 steps as defined in the *finetune_llff.txt* configuration. The training and evaluation is performed in the same $800 \times 600$ pixel resolution as for our own method.

**IDR.** We use the IDR implementation provided by the authors [1]. The training and evaluation is performed with the same training and test views and the same $800 \times 600$ pixel resolution as for our own method. We use the configuration for the DTU dataset provided by the authors which trains each scene for 2000 epoch where each epoch samples one batch of rays once for each of the seven training input views. This yields 14,000 solver steps for our input scenario.

**NLR.** We received access to the NLR implementation from the authors [3], and have used this for training comparison models. The training and evaluation is performed with the same training and test views and the same $800 \times 600$ pixel resolution as for our own method. The converged models were sampled after 100,000 solver steps, at which point loss curves stopped decreasing.

**SVS*.** We use our own mesh-based method implementation inspired by SVS [4], as described in the text. We use COLMAP 3.6 [13, 14] to reconstruct surface mesh from the subset of 7 training views with known intrinsic parameters for each of our test DTU scenes [2]. The input images are resampled to the same $800 \times 600$ pixel resolution as for training of our own method. We follow the settings described in previous work [1–3]. We remove background points from the fused point-cloud using the object masks and we performed Poisson reconstruction of the surface with trim parameter set to 7. For training the encoder, decoder, and aggregation function $E_\xi, D_\psi, \Gamma_\zeta$, we use the same parameters and architecture as NLR++. However, since no shape optimization is done, every iteration optimizes the feature processing networks.

**MetaNLR.** We received access to the NLR implementation from the authors [3], and have wrapped this implementation with an outer loop which learns the initialization of the shape and color networks using Reptile [10]. The meta-training and testing are performed with the same meta-training and meta-test scenes, and the specialization to the test scenes is done using the same training and testing views (at $800 \times 600$ resolution) as in our method. We independently tune the hyperparameter for meta-learning rate, which we set as $5 \times 10^{-2}$, and use the same amount of inner loop steps per meta-iteration as our method.

# 3 Additional Results

In this section we provide additional results of our method, and other baseline methods for various scenes.

## 3.1 DTU dataset

**Additional image renderings & SVS* results.** In our attached video and Figure 1, we show additional results comparing novel view synthesis results from various methods. In the video, we see that after a short amount of time, our method produces the highest quality novel view synthesis results. Note that as mentioned in the main text, the SVS* method's ability to synthesize novel views is limited by the inaccurate rendering masks in the COLMAP-produced geometry. This results in large holes in the rendered images. This phenomenon is less observed when quantitatively comparing rendered images using the ground-truth masks, as in the main text, but when rendering views without ground truth object masks, either the full image or rendered mask must be used to display images.

Note that the results in the video for SVS* also exhibit flickering artifacts between views, which are not apparent in the results produced by the original SVS implementation [4]. However, in the experiments presented, the number of training views is limited to only 7, and thus the feature processing networks $(E_\xi, D_\psi, \Gamma_\zeta)$ are more prone to overfitting to the training views. This overfitting results in less smooth blending between the training views. Additionally, the original SVS implementation uses a graph attention network instead of a MLP for on-surface feature aggregation, which may provide smoother feature interpolation.

We also include IBRNet and MetaNLR in these comparisons. IBRNet method is able to blend input images together and produce qualitatively good results, but cannot be rendered in near real-time. MetaNLR improves upon the training speed of NLR, but is still not able to match the visual quality of MetaNLR++ in the case of fast training, demonstrating that both our novel scene parameterization and the application of meta learning are necessary for fast training of high quality view synthesis methods.

**Ablation results & shape quality evaluation.** In Figure 2, we provide additional image results from the meta learning ablation study. We show qualitative results from meta-learned initializations specialized for 10 minutes of training. We show permutations of MetaNLR++ with RGB pixels instead of an encoder and decoder, and with a learned and fixed aggregation function. We see that the learned aggregation function introduces distortion in training when used with RGB pixel values directly, but leads to improved results when used with deep feature blending. The methods which use RGB pixel values directly are unable to inpaint and correct errors in the geometry, and thus lead to missing colored areas around the object boundary and falsely blended or occluded features when the shape is not correct.

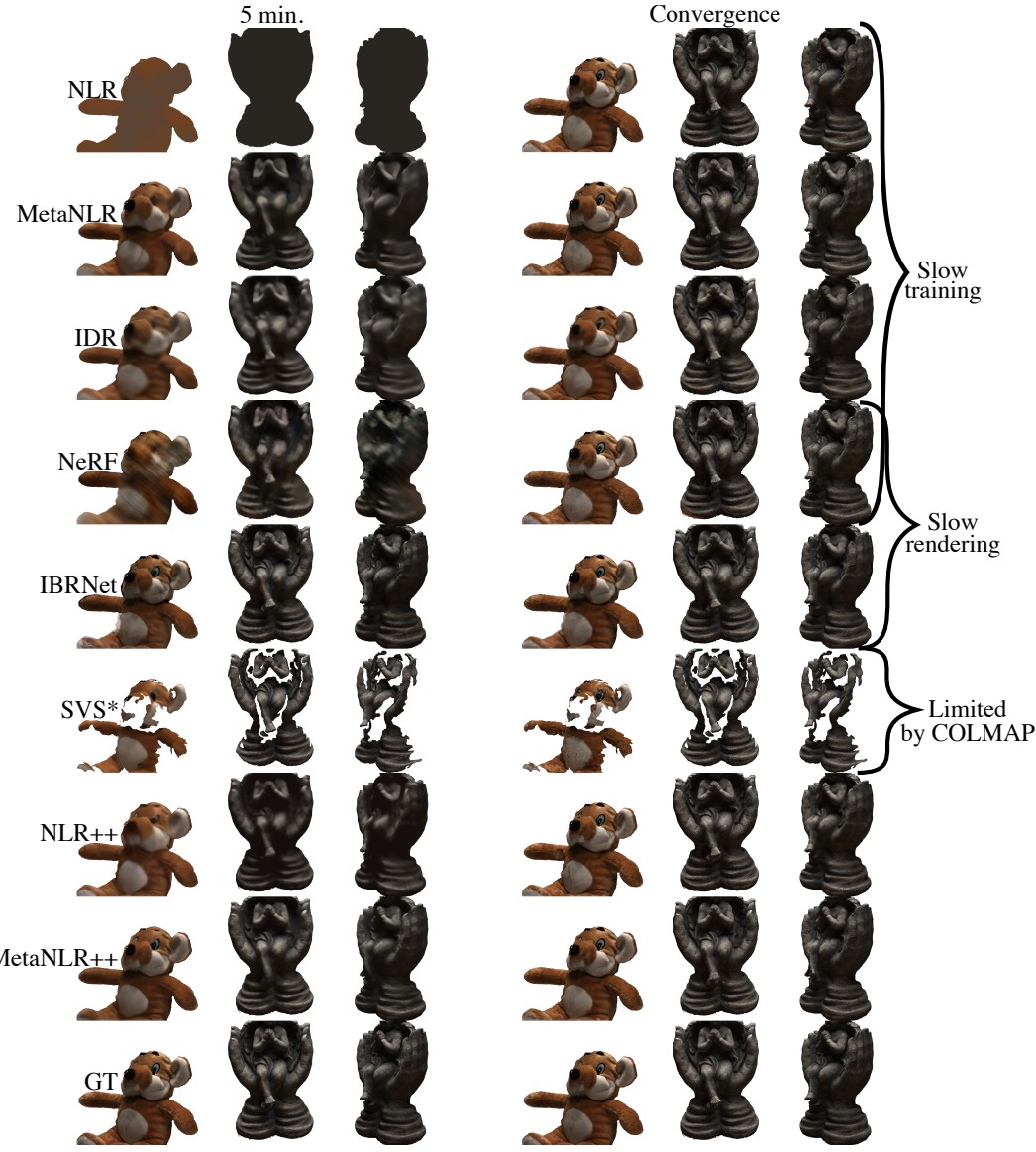

Figure 1: Additional qualitative results for all methods compared after 5 minutes of training and convergence. Here we apply the ground-truth background masks to all methods. Additionally, in the case of surface based methods, we also render test views using the masks extracted from the surface. This shows that the COLMAP results are limited by holes, especially in the mouse scene (DTU 105).

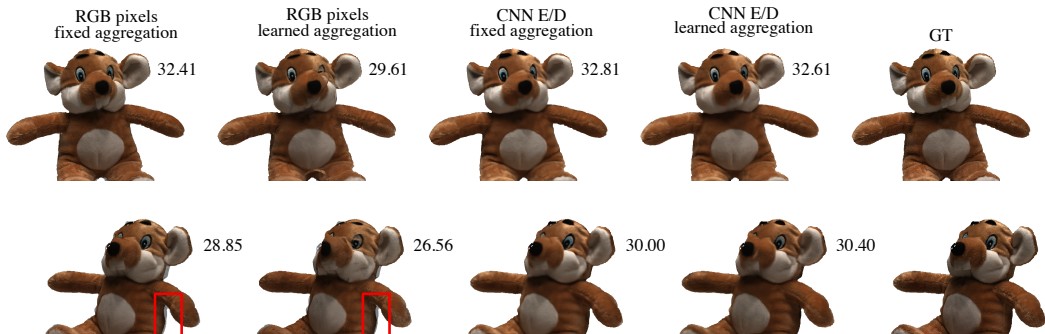

Figure 2: Comparison of the various ablated methods. In this comparison, we apply meta learning to all available networks, and show results after 10 minutes of training. The methods which use the RGB pixels produce sharp results, as the encoder and decoder do not need to be fine-tuned, but are not robust to errors in the geometry, as highlighted in the figure. These areas are inpainted in the version with a decoder.

|  | 5 min. Chamfer | 10 min. Chamfer | Convergence Chamfer |
|---|---|---|---|
| NeRF | 108.22* | 61.92 | *1.54* |
| IDR | **3.10** | **1.78** | **1.42** |
| NLR | 7.46 | 6.27 | 1.68 |
| COLMAP | 5.61 | 5.61 | 5.61 |
| NLR++ | 3.86 | 3.39 | 2.70 |
| MetaNLR++ | *3.26* | *2.78* | 2.28 |

Table 1: Table comparing Chamfer distances of the shapes produced from different methods at different times. These Chamfer results are averaged over the three DTU test scenes.
*For one of the DTU test scenes, a mesh was unable to be extracted, so this distance is the average of the other two scenes.

In Table 1, we compute the Chamfer distance for the methods shown in the main text. IDR performs well in reconstructing a high quality shape quickly, but as shown in the main text and Figure 1, rendered novel view synthesis is low-quality. This detail in shape is likely a result of the low capacity of the color representation, forcing the neural shape model to model differences in rendered views with geometry. MetaNLR++ achieves the second fastest shape convergence, significantly faster than volumetric methods such as NeRF, and faster than other neural surface based methods like NLR.

### 3.2 NLR dataset

For additional comparisons and convergence on the NLR dataset, please see our supplemental video.

**Failure cases.** As noted in the implementation details, when working with the NLR dataset, we increase the capacity of the neural shape representation and decrease the capacity of the feature decoder. Specifically, we increase the neural shape model $\Phi_\theta$ hidden dimension from $128$ to $256$ and decrease the number of UNet blocks in the decoder module $D_\psi$ from 3 to 2, with half the number of channels in each block. Without these architecture changes, the representation is unable to represent fine geometric details, and thus uses the decoder to decode the falsely blended features and inpaint the falsely occluded features to produce the training images. This results in overfitting to the training views. As shown in Figure 3, this is especially effective around the nose area of the face in the NLR dataset. Features here are blended when they should be occluded by the nose, resulting in blurriness. This blurriness is especially apparent in the interpolated views, which the decoder has not seen and thus cannot memorize the ground truth image.

### 3.3 ShapeNet dataset

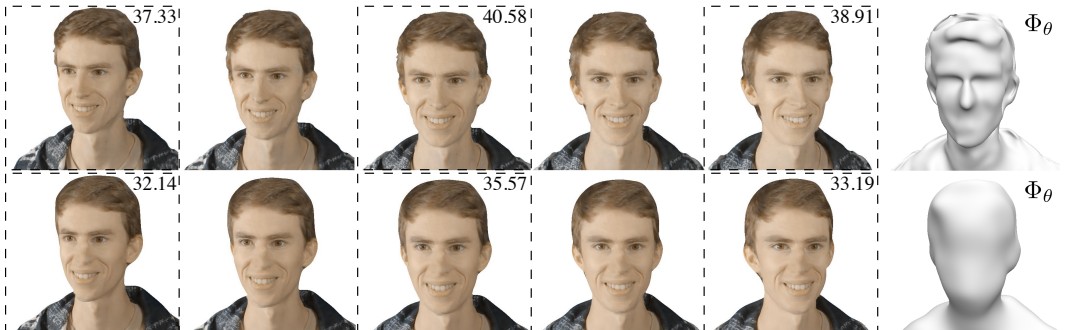

Figure 3: The first row shows the converged results on the NLR dataset with the architecture changes to $\Phi_\theta$ and $D_\psi$. The second row is a failure case, where the novel view synthesis quality interpolates poorly and is unable to fit the ground truth frames as well. This occurs because the shape model $\Phi_\theta$ does not have enough capacity to model the fine details of the face, and the $E_\xi, D_\psi, \Gamma_\zeta$ have too much capacity to overfit to training views.

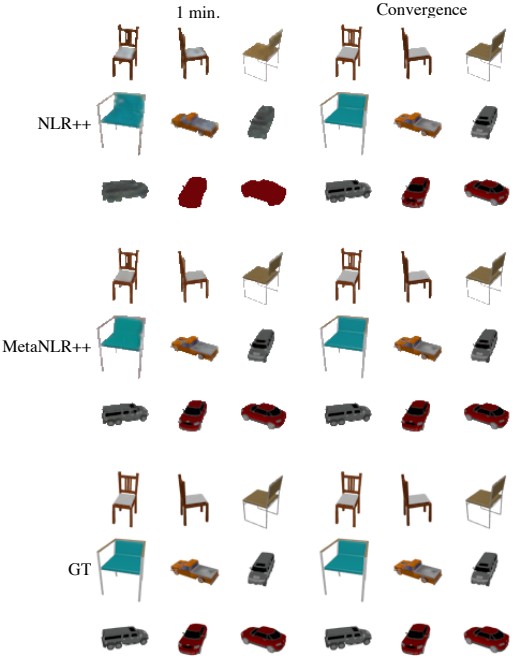

Figure 4: Qualitative examples from the cars and chairs splits of the ShapeNet dataset. These are shown after 1 minute of training and convergence for MetaNLR++ and NLR++.

We ablate the meta learning contribution by comparing MetaNLR++ to NLR++ on the cars and chairs splits of the ShapeNet dataset [15]. The relative performances are demonstrated in Table 2 by quantifying the time to reach a certain PSNR, and qualitative results are shown in Figure 4 after varying amounts of training. The testing and training views of the ShapeNet objects are distributed 360° around each object, demonstrating that MetaNLR++ is able to function well in this scenario.

From quantitative and qualitative results, we see that meta learning used in MetaNLR++ leads to a significant improvement over simply using the NLR++ scene parameterization. This continues the trend as described on the NLR dataset in the main paper, where a more comprehensive meta-training object dataset covering the testing object distribution leads to improved relative benefit of meta-learning. In the case of the cars split of the ShapeNet dataset, the lower variation between car objects than, for example, DTU objects leads to a stronger meta-prior learned and thus relatively faster training of MetaNLR++ compared to NLR++. This trend is observed to a lesser extent for the chairs split, where the objects are not as uniform as cars. We hypothesize that for future applications of MetaNLR++, larger meta-training datasets and specific applications could lead to even faster training.

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

| Cars Split | 30dB PSNR | 35dB PSNR | Maximum PSNR |
|---|---|---|---|
| NLR++ | 1.5 min. | 36.0 min. | 35.5dB |
| MetaNLR++ | **12.0 sec.** | **3.7 min.** | **37.9dB** |

| Chairs Split | 30dB PSNR | 35dB PSNR | Maximum PSNR |
|---|---|---|---|
| NLR++ | 3.6 min. | 125.0 min. | 37.0dB |
| MetaNLR++ | **1.5 min.** | **50.0 min.** | **40.7dB** |

Table 2: Comparison of the time to reach a specified PSNR level on the ShapeNet dataset for MetaNLR++ and NLR++.

[4] Gernot Riegler and Vladlen Koltun. Stable view synthesis. In *CVPR*, 2021.

[5] Kaiming He, Xiangyu Zhang, Shaoqing Ren, and Jian Sun. Deep residual learning for image recognition. In *CVPR*, 2016.

[6] Olaf Ronneberger, Philipp Fischer, and Thomas Brox. U-net: Convolutional networks for biomedical image segmentation. In *Medical Image Computing and Computer-Assisted Intervention – MICCAI 2015*, pages 234–241, 2015.

[7] A. Dosovitskiy, P. Fischer, E. Ilg, P. Häusser, C. Hazırbaş, V. Golkov, P. v.d. Smagt, D. Cremers, and T. Brox. Flownet: Learning optical flow with convolutional networks. In *IEEE International Conference on Computer Vision (ICCV)*, 2015. URL `http://lmb.informatik.uni-freiburg.de/Publications/2015/DFIB15`.

[8] E. Ilg, T. Saikia, M. Keuper, and T. Brox. Occlusions, motion and depth boundaries with a generic network for disparity, optical flow or scene flow estimation. In *European Conference on Computer Vision (ECCV)*, 2018. URL `http://lmb.informatik.uni-freiburg.de/Publications/2018/ISKB18`.

[9] Diederik P Kingma and Jimmy Ba. Adam: A method for stochastic optimization. *ICLR*, 2014.

[10] Alex Nichol, Joshua Achiam, and John Schulman. On first-order meta-learning algorithms. *arXiv preprint arXiv:1803.02999*, 2018.

[11] Ben Mildenhall, Pratul P. Srinivasan, Matthew Tancik, Jonathan T. Barron, Ravi Ramamoorthi, and Ren Ng. Nerf: Representing scenes as neural radiance fields for view synthesis. In *ECCV*, 2020.

[12] Qianqian Wang, Zhicheng Wang, Kyle Genova, Pratul Srinivasan, Howard Zhou, Jonathan T. Barron, Ricardo Martin-Brualla, Noah Snavely, and Thomas Funkhouser. Ibrnet: Learning multi-view image-based rendering. In *CVPR*, 2021.

[13] Johannes Lutz Schönberger, Enliang Zheng, Marc Pollefeys, and Jan-Michael Frahm. Pixelwise view selection for unstructured multi-view stereo. In *ECCV*, 2016.

[14] Johannes Lutz Schönberger and Jan-Michael Frahm. Structure-from-motion revisited. In *CVPR*, 2016.

[15] Angel X. Chang, Thomas Funkhouser, Leonidas Guibas, Pat Hanrahan, Qixing Huang, Zimo Li, Silvio Savarese, Manolis Savva, Shuran Song, Hao Su, Jianxiong Xiao, Li Yi, and Fisher Yu. ShapeNet: An Information-Rich 3D Model Repository. *arXiv preprint arXiv:1512.03012*, 2015.