# OpenReview forum: "Fast Training of Neural Lumigraph Representations using Meta Learning"
_NeurIPS.cc/2021/Conference — NeurIPS 2021 Poster_

### Official Review · Reviewer_cxjr · 2021-07-15

**Rating:** 5
**Confidence:** 2

**Summary:**

The paper finds a set of weight values that is a good initialization to train a recent novel-view synthesis (NVS) method, Neural Lumigraph Rendering (NLR). A meta-learning library, Reptile, is used to optimize this initial weight vector across a corpus, the DTU multiview dataset. The paper shows, that with such an initialization, optimization proceeds much quicker and to similar quality. Additionally, it is suggested to change NLR to aggregate features from the input images instead of pixel values directly, a change that appears to be orthogonal to the meta-learning aspect. Results show that after much fewer optimization steps, a good result is achieved. The method is fast, but that is a property of NLR.

**Ethical Concerns:**

No idea

**Ethics Review Area:**

["I don’t know"]

**Limitations And Societal Impact:**

No idea

**Main Review:**

The topic of meta-learning is original, yet the same idea of finding a good init has now been applied to the novel-view synthesis based on overfit neural networks twice, once by Sitzman et al. and once by Tancik. et al. The difference between all three approaches is mostly that they are applied to different backbone NVS methods, but the use of Reptile itself to tune the init is the same. Hence the topic is novel, but within this new trend, this approach is not adding much new to the two existing papers, besides the insight that what worked for DeepSDF and for NeRF word for NLR, too. This might not be enough.

The quality of the result and experiments is high. All experiments are performed very convincingly and systematic. One could imagine comparing to Tancik et al. They do metalearning for NeRF, this does metalearning to NLR. Both can be compared on a novel-view task after N iterations as done in Fig. 3 with non-metalearned approaches.

The writing is exceptionally clear, modulo that what is described is not always very interesting in the sense that it does not matter so much to the main topic. Most readers will want to learn something interesting about metalearning, but it boils down to run Reptile and a mere init vector optimization. A good deal of the paper is about a change from pixel aggregation to feature aggregation, and also this is not quite interesting and had been done in many papers before. it really only is NLR++, an incremental change to NLR, inspired by many other recent papers.

The problem is significant, as training times are a large issue and quick adaption is a much desired goal.

Overall, reading the paper left this reviewer slightly disappointed. A first brief look is very promising. A second, skim, read identified three relevant takeaways: 1) Reptile applied to two NVS methods before can be applied to NLR, too 2) Aggregating features is better than aggregating pixels, also shown for many NVS works before 3) NLR++ is high-quality, fast and learned quickly. A third, detailed, read did not bring anything substantial to note, resulting in a borderline assessment.

**Time Spent Reviewing:**

2

---

> ### Author Response · Authors · 2021-08-09
> **Response to Reviewer cxjr**
>
> We thank Reviewer cxjr for their time spent on reviewing and providing a detailed and thoughtful review which will certainly help us improve our work. We appreciate the comments of the reviewer which describe the clarity of the writing and quality of the experiments, and certainly agree that fast training and rendering is a very significant problem for neural scene representations. MetaNLR++ is the first method to address fast training, and simultaneously also addresses fast rendering. We address the reviewer’s individual concerns below:
>
> **To clarify the relationship between this work and NLR**, it is not accurate that NLR aggregates pixels on the surface of some geometry representation. NLR trains two coordinate-based networks: one which represents the object surface (like in NLR++), and one which represents color densely in 3D (not on the surface, unlike our method). Thus, there is no pixel aggregation or use of the input images in general beyond training this network. Our paper observes that training a network to represent color densely in 3D results in significantly longer training times, as every time the shape representation is updated, the color representation must also update to correctly reflect the appearance on the modified surface of the shape. We will make this distinction and fundamental limitation of NLR more clear in the paper. Following this analysis, we agree that the fast rendering is a property of NLR, but fast training is absolutely not, as shown in Table 1, Figure 2, and Figure 3. To this end, NLR++ proposes to replace this color representation network with features extracted from the input images. As for pixel aggregation versus extracted feature aggregation, we show an ablation in Table 2 and Supplement Figure 2, where the pixel values are taken instead of features, and demonstrate that, as the reviewer notes, features improve the synthesized image quality. Furthermore, even the simplified version of NLR++ directly using the pixel values is novel on its own, as no other method aggregates any input image values (features or pixels) on the surface of a learned neural shape representation, as noted by the Reviewer UVXv and Reviewer LYqQ . We will be sure to add this needed explanation regarding pixel versus feature aggregation in the paper, where the pixel aggregation method is ablated.
>
> Additionally, following this clarification of the difference between NLR++ and NLR, we can also show that **simply applying meta learning to NLR does not lead to nearly as fast of training to high quality when compared to MetaNLR++**. Hence, both the improvements in the forward model by using aggregated features, and the meta learning, are necessary components that jointly contribute to achieving the desired goal of fast training, while preserving the fast rendering of NLR. To demonstrate this, consider the entries to Table 1 of the NLR method using meta learning, where the results for MetaNLR++ are extracted from the paper for comparison:
>
> | Method              | Time to reach 25dB | Time to reach 30dB | Maximum PSNR       |
> |---------------------|--------------------|--------------------|--------------------|
> | Meta learning + NLR | 2.1 min            | 176.8 min          | **31.71dB (1105 min)** |
> | MetaNLR++           | **1.9 min**            | **22.5 min**           | 30.57dB (125 min)  |
>
> We will add this baseline result to Table 1 and qualitative results to the Supplement, which show that MetaNLR++ is significantly faster at reaching the 30dB PSNR baseline than simply applying meta learning to the NLR method.
>
> We really appreciate that the reviewer finds our result quality high, and the analysis of our method convincing and systematic. While it would be very interesting to compare to an adapted version of [1], in the prior literature, meta learning for volume rendering models has only been demonstrated for the purpose of generalization over varying environment conditions of the same scene and not for fast training. Thus, it may be beyond the scope of this paper to adapt [1] for fast training. This is also true of [2], which demonstrates the use of meta learning for learning a prior over a class of signals, but doesn’t specifically optimize for fast training of complex representations. While all of these works share a similar method for generalization (meta learning), the representation which is this is applied to (volume model, SDF representations) make a large difference in the possible applications, such as novel view synthesis or shape representation, and properties, such as training and rendering speed and quality.
>
> [1] Matthew Tancik, Ben Mildenhall, Terrance Wang, Divi Schmidt, Pratul P Srinivasan, Jonathan T Barron, and Ren Ng. “Learned Initializations for Optimizing Coordinate-Based Neural Representations”. In CVPR, 2021.
>
> [2] Vincent Sitzmann, Eric R. Chan, Richard Tucker, Noah Snavely, Gordon Wetzstein. “MetaSDF: Meta-learning Signed Distance Functions”. In NeurIPS, 2020.

---

### Official Review · Reviewer_LYqQ · 2021-07-16

**Rating:** 6
**Confidence:** 4

**Summary:**

This paper presents a method for capturing the 3D shape and appearance of objects using an implicit model for the shape, and an aggregation of reprojected CNN features for the appearance. The authors highlight two problems with existing approaches that they try to address - slow training and rendering times. The key contribution of the paper is the use of meta-learning, in the form of a learned initialization, to improve the efficiency of the training process. The results show that the proposed method achieves a marginal improvement in training efficiency.

**Limitations And Societal Impact:**

The paper does a good job at describing the potential negative societal impacts.

**Main Review:**

**TL;DR:** A useful idea to speed up the training and rendering of neural scenes, but the practical improvements are not that impressive. In the main review I have labelled points as a strength (+), weakness (-) or mixed (+/-).

**Originality**:

(+) The method is new as I am not aware of any existing work that uses a combination of neural level-set surfaces and aggregated CNN features for appearance. There are, however, existing works that have investigated the use of learned initializations for training NeRF-like models but these have not been used in combination with generalizable features from a CNN.

(+) Although all of the components of the approach (the SDF representation, the CNN features and the learning initialization) are all well-known techniques, the combination is novel and valuable as they all work together to improve the training efficiency.
The authors have done a good job at differentiating this work from previous contributions (such as MetaSDF and [78]) and, as far as I can tell, have adequately cited all the related work.

**Quality**:

(+) The submission appears to be technically sound and the details of the method are well documented. One technicality that I am not so sure about is why in Section 3.2, out of the k images that are sampled, only one is used as the ground-truth target image. Surely all of the selected images can act as training targets?

(+/-) The claims of improved training efficiency and quality are supported by the quantitative evaluation that compares the approach to recent methods like SVS, IBRNet, NLR. However, the improvements in training time and PSNR are not that major (see significance below).

(+) Overall, the paper is definitely a complete piece of work. The method is fully developed and it has been evaluated and compared to recent approaches on two different datasets. The figures, results and writing are well polished.

(+) The approach has a number of weaknesses, such as the need to use object silhouettes as input (and therefore can only reconstruct single objects and not the whole scene). Another disadvantage is that the reconstructed geometry is rather coarse and not very detailed. However, the authors have done a very good job at explaining and justifying these limitations and providing ways in which they could be addressed in future work.

**Clarity**:

(+) The submission is very clearly written and the overall structure of the paper is well organized. As far as I can tell the paper provides enough information about the method and the experimental setup for a reader to be able to reproduce its results.

**Significance**:

(-) The results are important in the sense that fast training and rendering of neural scene representations is challenging and an open problem. This paper addresses the training speed in two ways - through the use of generalizable features extracted from a CNN and by using meta-learning, where the main contribution seems to be the use of the meta learning (as per the title). Although it is conceptually a good idea, the real improvements gained by the meta learning seem to not be that significant - for example 27.32dB vs 29.91dB in the best case in Table 2. Is this difference even perceptually significant? Are there ways in which the meta learning could be improved to get a more significant gain in PSNR?

(+/-) There are ways in which others are likely to use or build upon the ideas presented in the paper, but the seemingly minor improvement in the training speed and accuracy might dissuade many in doing so.

(+) The method does advance the state of the art in terms of training speed (albeit in a very slight manner). In terms of this comparison I would say that IBRNet should be included in Table 1 as there are ways in which it could be be turned into a pre-computed volume representation (for example, by storing the projected features from the nearest images in an Octree).

# Post response and discussion
After considering the authors' response and the other reviews, most of my concerns have been addressed. I agree with rL4a that the core components of the proposed method come from elsewhere and there does not seem to be any special theoretical contributions to these in the proposed paper (most are used as is). However, I think the approach is well-engineered and it seems to achieve good results in terms of training efficiency vs quality which is an important problem for these types of neural reconstruction+rendering approaches.  My final rating is therefore "6: Marginally above the acceptance threshold".



**Time Spent Reviewing:**

2

---

> ### Author Response · Authors · 2021-08-09
> **Response to Reviewer LYqQ**
>
> We thank Reviewer LYqQ for reviewing and providing a thoughtful and extensive analysis of our work.
>
> We appreciate the positive comments recognizing the method originality and clarity and quality of the writing, figures, and results. We especially appreciate the recognition that the method is a novel combination of neural level-set surfaces and image feature aggregation which uses learned initializations, and is distinct from previous work in meta learning and neural rendering. Regarding the technical confusion about the number of target images at each iteration, the reviewer is correct, and some subset $l<k$ of the source images is selected to be training targets, instead of only a single image. The value of $l$ is limited by the memory of the GPU when computing gradients with respect to the target image loss. We will be sure to clarify this in the paper.
>
> **Regarding the limitations of the approach**, the reviewer is absolutely correct in noting that our method requires the usage of object silhouette masks as input, and the reconstructed geometry is smooth and not very detailed. As the reviewer mentions, we extensively discuss these limitations in the paper, and will continue to stress that the core ideas of meta learning applied to a neural surface representation and working with on-surface aggregated image features are still applicable to concurrent methods which bootstrap neural surface model training using a volumetric model and remove the silhouette mask requirement, such as [1-3]. In addition, in Supplementary Figure 3, we show ablations which demonstrate that the level of smoothness / detail learned by the geometry representation is dependent on the capacity of the CNN feature encoders/decoders and neural network modelling the object surface. Since the goal for MetaNLR++ is to quickly learn representations amenable to novel view synthesis of images, and highly informative CNN features are necessary to allow the capacity for modeling sharp image details, we believe that the smooth geometry may only be a minor limitation for most applications of the method. Balancing this trade-off between geometry and color detail while still maintaining the fast training properties is an area that is ripe for future research, and we will emphasize this in the paper.
>
> **Regarding the significance of the improvements due to meta learning**, the reviewer is correct that our NLR++ alone is already able to perform remarkably well at fast training and significantly outperforms both NLR and NeRF (Table 1, Figure 2, Figure 3). However, our experiments show that the meta learning capability of MetaNLR++ is critical to achieve truly significant training speed improvement. Moreover, this extra margin grows for datasets where the meta-training domain well covers the test domain. In Table 3, we illustrate this on the NLR dataset, where all objects are human heads, by showing that meta learning leads to an increased benefit in synthesized image accuracy (PSNR) and a 1.75x increase in image perceptual quality (as measured by the LPIPS metric) after a short amount of training. This pattern is even stronger in our experiment with the ShapeNet dataset (see table in response to the Reviewer UVXv). We will discuss this insight in the paper as it is useful for future work and applications of our method with larger, higher quality, meta-training datasets.
>
> Beyond fast training, close inspection of Figure 3 demonstrates that the results from MetaNLR++ are often notably less blurry than NLR++, exemplifying the improved performance due to meta learning. We will make this qualitative difference more clear in Figure 3.
>
> Finally, we agree and note that IBRNet could be modified (although possibly not trivially) to a pre-computed volume representation. We will include the reported numbers for IBRNet (line 249-250) in Table 1, along with the rest of the methods.
>
> [1] Michael Oechsle, Songyou Peng, Andreas Geiger. “UNISURF: Unifying Neural Implicit Surfaces and Radiance Fields for Multi-View Reconstruction”. In ICCV, 2021.
>
> [2] Lior Yariv, Jiatao Gu, Yoni Kasten, Yaron Lipman. “Volume Rendering of Neural Implicit Surfaces”. arXiv preprint arXiv:2106.12052, 2021.
>
> [3] Peng Wang, Lingjie Liu, Yuan Liu, Christian Theobalt, Taku Komura, Wenping Wang. “NeuS: Learning Neural Implicit Surfaces by Volume Rendering for Multi-view Reconstruction”. arXiv preprint arXiv:2106.10689, 2021.

---

> > ### Comment · Reviewer_LYqQ · 2021-09-03
> > **Re: Response to Reviewer LYqQ**
> >
> > Thanks for the detailed response, it has clarified most of the issues I raised, particularly related to the number of images used for the training targets, the limitations of the approach and the significance of the meta learning. Although most of the ideas in the paper have been introduced elsewhere, I like the combination of the aggregated image features for appearance and the neural level set surface for representing the geometry and the meta learning provides a further boost in the performance. Overall, it is a well engineered approach.

---

### Official Review · Reviewer_UVXv · 2021-07-17

**Rating:** 8
**Confidence:** 5

**Summary:**

This paper presents MetaNLR++, a novel method for representing and rendering novel views of object-centric scenes. The scene representation combines a CNN-based feature extractor which extracts deep features from input views and a Neural SDF module which serves as a proxy geometry surface for transforming the extracted features to the novel views. The transformed features are then aggregated and processed by a CNN-based decoder to synthesize the output image. This use of image-based and surface-based scene representation enables faster rendering time compared to volume-based representation such as NeRF. A meta-learning technique based on learned initialization was leveraged to enable fast fitting of the model to new testing scenes. Experiments on two novel-view synthesis benchmarks demonstrate that the proposed method performs favorably compared to existing methods while requiring significantly less time for convergence.

**Limitations And Societal Impact:**

The limitation and impacts have been discussed clearly in the paper.

**Main Review:**


Strength:
 + The paper is well-written and easy to follow.
 + The proposed method is novel. It effectively addresses two important problems in novel-view synthesis: 1) Constructing the scene representation that enables fast rendering, and 2) Fitting such representation for novel scenes at test time in an efficient manner. The proposed method addresses (1) by combining classical image-based rendering pipeline (using a geometry proxy for transferring visual content from input views to novel views) with neural implicit function model for representing shapes and rendering. To address (2), a meta-learning technique is used which enables fast convergence time for fitting to a test scene.

Weaknesses:
 + There is no serious weakness from my view. However, I feel that more discussions can be added to help the readers better connect this work to existing knowledge in novel-view synthesis and image-based rendering. For example, it can be noted that the proposed method leverages a standard pipeline in image-based rendering that transfer visual information from input views to target views using a proxy shape, and it can be emphasized that the proxy shape in this case is a learned neural implicit function which is the novel contribution of this work. Also, the idea of using CNN-based features instead of color values to represent the information to be transferred from the input views has also been leveraged before for view synthesis. For example, [1] transfers deep features from a single input view to the target view using a learned depth map. I believe the readers can benefit from more discussions on the connection between this work and those existing works.
 + The idea of using meta-learning to address the training-time problem makes sense. The proposed method was shown effective in the experiments using NLR and DUT dataset. The number of scenes in those datasets, however, is relatively small. I'm curious how will different properties of the training data distribution (e.g. number of training scenes, similarity between the training scenes and the testing scene) affect the effectiveness of the meta learning step. I believe synthetic datasets such as ShapeNet can be used to design more controlled experiments to investigate such aspects.

[1] Olivia Wiles, Georgia Gkioxari, Richard Szeliski, Justin Johnson. "SynSin: End-to-End View Synthesis From a Single Image". Proceedings of the IEEE/CVF Conference on Computer Vision and Pattern Recognition (CVPR), 2020.

**Time Spent Reviewing:**

3

---

> ### Author Response · Authors · 2021-08-09
> **Response to Reviewer UVXv**
>
> We thank Reviewer UVXv for spending time on reviewing and providing insightful comments regarding our paper - they will certainly help improve the paper’s quality.
>
> We appreciate the reviewer acknowledging the clarity of presentation and novelty of our method, which addresses two problems in novel view synthesis by proposing two complementary contributions: a scene representation which enables fast rendering and a generalization method for fitting representations of new scenes quickly.
>
> We will expand our discussion on the **relationship between this method and existing novel view synthesis and image-based rendering methods**. We will make clear the distinction that image-based rendering methods either require estimating a proxy shape using a pre-processing step (like COLMAP, in the case of SVS [1]), or require the proxy geometry to be very simple (such as a sphere), while MetaNLR++ allows for end-to-end optimization of the shape using a coordinate-based network representation. We will also include discussion on SynSin [2], which also similarly projects extracted image features between viewpoints. However, SynSin leverages a point cloud geometry representation constructed from a monocular depth estimate, and only is applied to single-image novel view synthesis, and thus does not require any feature aggregation nor direct geometry representation unlike our method and SVS.
>
> We certainly agree with the sentiment that performing **ablations on the quality of the meta-learned initialization** with respect to the size and uniformity of the training dataset would be very insightful, and could likely lead to even further improvements of the meta learning approach. This is supported by our ablations between MetaNLR++ and NLR++ on the DTU and NLR datasets showing that in the case where the objects are more similar to each other (in the NLR dataset, all objects are human heads), MetaNLR++ has a larger relative benefit over NLR++ (Table 3). We have run a similar comparison between MetaNLR++ and NLR++ for objects in the ShapeNet dataset in the cars class, which confirms this trend and shows that, as the reviewer mentions, the additional training data and training versus testing distribution has a large effect on the importance of the meta learning aspect.
>
> | ShapeNet Cars | Time to reach 30dB | Time to reach 35dB | Maximum PSNR     |
> |---------------|--------------------|--------------------|------------------|
> | NLR++         | 1.5 min            | 36.0 min           | 35.5dB (100 min) |
> | MetaNLR++     | **12.0 sec**           | **3.7 min**            | **37.9dB (100 min)** |
>
> We will make the insight from this ablation clear in the paper, as we hypothesize that for future applications of our method, this could lead to even stronger meta-learned priors and faster training in many applications. We will include the quantitative ShapeNet results provided here, and qualitative results in the Supplementary information as we agree with the reviewer that it provides a very valuable insight.
>
> [1] Gernot Riegler and Vladlen Koltun. “Stable View Synthesis”. In CVPR, 2021.
>
> [2] Olivia Wiles, Georgia Gkioxari, Richard Szeliski, Justin Johnson. "SynSin: End-to-End View Synthesis From a Single Image". In CVPR, 2020.

---

### Official Review · Reviewer_rL4a · 2021-08-05

**Rating:** 3
**Confidence:** 4

**Summary:**

This paper presents an approach to improving the training performance and overall quality of the recently introduced Neural Lumigraph Rendering (NLR) framework for novel view synthesis for objects, using a meta-learning framework that allows for quickly computing a representation of new scenes that can be rendered in real time.

The framework encodes the object surface using a neural signed distance function (SDF) representation of the surface, which can then be efficiently decoded using sphere tracing techniques. The framework encodes features from the 2D input images, which are aggregated from images that are visible in the target and decoded into the reconstructed target view. View-dependent effects are achieved using an MLP that acts as a learned weight-mapping function for the features aggregated from each input view and processed by the decoder. This decoder, in processing these feature maps, can fix artifacts caused by imperfect geometry reconstruction in the surface aggregation function. The Reptile algorithm is used as part of a meta-learning framework to learn a good parameter initialization from multi-view datasets that allows for quickly converging to a representation for previously unseen objects. Evaluations suggest that this approach achieves a good balance between fast convergence on unseen objects, final image quality and fast rendering speed compared to recent alternatives including the original NLR.

Overall my impression is that the main contributions of this work are in the approach described to use more sophisticated shape and appearance representation compared to the original NLR, using image encoders and decoders and a more sophisticated weight-mapping function to achieve superior results while allowing for comparable inference speeds after training. My impression is that the meta-learning framework as described, which is largely based on the Reptile algorithm, is not as substantial a contribution.

**Ethical Concerns:**

There are no particular ethical concerns about this paper.


**Limitations And Societal Impact:**

Some broad limitations are discussed, although I think that some of the additional points raised in the main review above should also be addressed.

There are no particular issues w/ potentially negative societal impact that need to be addressed.


**Main Review:**

Overall this work presents an approach that seems reasonable and attains reasonably good results. The clarity of exposition is generally quite good, and the supplemental information and video help to illustrate the strengths and limitations of this approach. However, I am not inclined to argue in favor of acceptance due to some key concerns that I have:

- I noticed that for this paper (and the prior Neural Lumigraph Rendering work it extends), all of the depicted results appear to be for synthesizing novel views from a limited range of mostly frontal input images, similar to those from LLFF dataset used by NeRF. However, NeRF also demonstrates reasonable results on full rotations around isolated objects, e.g. synthetically rendered objects for which arbitrary input training views sampled from a sphere or hemisphere around the target are available. Given that this approach appears to be targeted at isolated foreground objects, I'm curious whether this approach would work for a similar range of novel viewpoints, given appropriate training data.

- While the qualitative and quantitative results for this work appear to validate the authors' claims, and the efficient inference performance is an attractive trait of this work, its apparent limitation to segmented foreground objects is a fairly strong limitation compared to the other evaluated approaches that can also be applied to images of complete scenes, e.g. the NeRF results on LLFF image sets.

- The core approach of this work is largely derived from previous work, which limits the overall novelty of the contributions. The extensions to this approach seem reasonable, but many are also relatively straightforward, e.g. using a network pre-trained on large object datasets to allow for faster adaptation to new objects, and using the Reptile algorithm as described in the paper for meta-learning.

This overall approach to meta-learning from other scenes appears to have some similarities to that of pixelNeRF, except that the latter uses a network trained on a large number of objects/scenes to allow for novel view synthesis using as little as 1 new image without any additional training. I am curious whether allowing for further fine-tuning of such a network given a set of multiple new images to obtain comparable results. I understand that this is a significant deviation from their approach, and don't expect a direct comparison to what I describe, but I would be curious to hear any comments from the authors or other reviewers on the relationship between these 2 approaches.

- As noted by the authors, the overly smooth nature of the results and the requirement of object masks to supervise the ray-mask loss are fairly strong limitations compared to other methods that produce high-quality results for complex, unmasked scenes with multiple objects.

- The SDF and sphere-tracing approach seems reasonable for opaque objects, and allows for efficient rendering of such objects compared to the volume rendering approach used by recent works using neural radiance fields, but it appears that its ability to account for complex effects such as translucent objects (e.g. glass) is limited. While this is an understandable tradeoff given the performance benefits, it should be noted as a limitation compared to some of the alternative approaches that are discussed and evaluated. In contrast, recent work like Nex [33] can handle such effects on full shiny/translucent while also maintaining real-time inference performance using an MPI-based approach, making me question the overall efficacy and value of this alternative approach.

Some minor points:

- I think that the notation used for the camera parameters, e.g. { C_i, R_i | t_i }, could be shortened to make the notation in the equations easier to read and follow.
- L231: emphesize -> emphasize


**Time Spent Reviewing:**

~4 hours

---

> ### Author Response · Authors · 2021-08-09
> **Response to Reviewer rL4a**
>
> We thank Reviewer rL4a for their time spent reviewing, and providing an in-depth review of our work. We appreciate the comments which recognize the more sophisticated shape and appearance representation proposed in NLR++, and the clarity of exposition which clearly demonstrate the strengths and limitations of the approach. We address the concerns brought up below:
>
> **In terms of the novelty of the approach and the contribution of the meta learning aspect**, we demonstrate that while meta learning, on-surface feature aggregation, and learned shape representations may have all been studied in independent work, no other work has combined these techniques in the same way as our framework nor attempted to approach the problem of fast training of these representations, as noted by Reviewers UVXv and LYqQ. Fast training and rendering are significant problems and currently one of the largest roadblocks in the widespread adoption of neural rendering techniques, and our work combines many aspects of neural rendering in a novel way to address this problem. The reviewer is correct that the paper proposes a more sophisticated shape and appearance representation compared to the original NLR as a contribution, but the prior encoded via meta learning is a key aspect of fast convergence, as demonstrated by the increased performance of MetaNLR++ in Table 1, 2, 3, and Figures 2, 3. Additionally, the meta learning is not an independent contribution from the representation; simply applying the Reptile algorithm to the method in NLR does not yield nearly as fast convergence results, as demonstrated by the following comparison:
>
> | Method              | Time to reach 25dB | Time to reach 30dB | Maximum PSNR       |
> |---------------------|--------------------|--------------------|--------------------|
> | Meta learning + NLR | 2.1 min            | 176.8 min          | **31.71dB (1105 min)** |
> | MetaNLR++           | **1.9 min**            | **22.5 min**           | 30.57dB (125 min)  |
>
> We will edit the paper to better communicate this relationship between the two main contributions (representation architecture and meta learning) and the desired application area, which is for the first time among neural rendering methods tackling simultaneously both fast training and fast rendering.
>
> **Regarding the limitations of the method**, we extensively discuss the object silhouette mask requirement and smooth geometry in the paper, as reviewer LYqQ notes. We will make it significantly more clear that the core ideas of meta learning applied to a neural surface representation, and working with on-surface aggregated image features are still applicable to concurrent methods in this active area of research which remove the silhouette mask requirement [1-3]. Similarly, we will emphasize the analysis presented in Supplementary Figure 3, which demonstrates that the smooth geometry is dependent on the capacity of the CNN feature encoder/decoder and neural network modelling the object surface. For the purpose of fast training for high quality image synthesis, as is the case in MetaNLR++, we trade off some of the shape representation network capacity into the appearance modelling image features, but finding a way to balance this ambiguity in appearance modelling between features and geometry while maintaining fast training properties is ripe for future research and developing methods which may build upon the ideas presented in this paper.
>
> **Regarding the limitations on input view distribution**, while it is true that this paper only shows results from views synthesized from a range of mostly frontal input images, there is no technical bias in the representation which limits learning representations of objects from input views distributed arbitrarily around the object. This is demonstrated in the original NLR work (which  NLR++ builds on) on the Volucap dataset, and we’ve provided additional results on the ShapeNet dataset as provided by the DVR paper (for the purpose of demonstrating the increase in relative improvement of meta learning when more training data is available) which distributes these views in a ring around the object. In the table below with ShapeNet results, we see that NLR++ and MetaNLR++ are still able to produce accurate view reconstructions on held-out views distributed 360 degrees around the object.
>
> | ShapeNet Cars | Time to reach 30dB | Time to reach 35dB | Maximum PSNR     |
> |---------------|--------------------|--------------------|------------------|
> | NLR++         | 1.5 min            | 36.0 min           | 35.5dB (100 min) |
> | MetaNLR++     | **12.0 sec**           | **3.7 min**            | **37.9dB (100 min)** |
>
> We will add these comparisons and analysis regarding a wider distribution of input views to the Supplementary information in the paper, as we agree that this is an interesting application area where our method is still capable of performing well.
>
> **Regarding the limitations of surface-based methods in general**, we certainly agree with the reviewer that this class of approaches has a number of limitations (e.g. modelling translucent objects) and a number of benefits (e.g. allowing for efficient rendering). We will certainly make our discussion about where MetaNLR++ falls within this class of methods more comprehensive by noting the drawbacks of these approaches as a whole. However, we believe that fast training and rendering of these representations is a very important (and depending on application, possibly the most important) issue limiting widespread adoption of neural rendering techniques, and as such the trade-offs are warranted. Recent work such as NeX [33] may be able to handle more complex viewing effects and render in real-time, but makes sacrifices in terms of fast training (the paper describes a single scene requiring on the order of 18 hours to train). While this may be the right approach to build on for some applications, the MetaNLR++ approach has a significant value in applications where surface based methods are desired and fast representation learning is crucial, such as capturing people-related content for teleconferencing or virtual reality. We will delineate this value more clearly in the text, which will more effectively communicate our motivation for choosing this kind of representation.
>
> While both **MetaNLR++ and PixelNeRF** share an aspect of generalization by learning a function which combines input image features in a meaningful way, they are different in representation style and application. MetaNLR++ explicitly models the surface and focuses on high-quality fast training, while PixelNeRF uses volume rendering and focuses on an extremely low number of input views. However, the results of PixelNeRF show that the quality of the results does not increase much with an increased number of views, which we hypothesize is due to the lack of explicit occlusion reasoning outside of the volumetric rendering method. A very related method, IBRNet [28], has been proposed which similarly uses a volumetric rendering method, but includes a transformer to explicitly reason about occlusion along each ray, producing higher quality results with no or little scene fine-tuning. We compare our work to IBRNet in Section 4 (training time vs quality trade-off, training to convergence), and explain that while it is capable of quickly reaching high-quality results based off of the prior learned in the feed-forward networks, because it is based on volume rendering, it sacrifices the fast rendering aspect that MetaNLR++, NLR++, and NLR all share. Thus, while the experiment described with fine tuning the PixelNeRF architecture for a novel scene is conceptually very similar to IBRNet and would likely be able to synthesize high-quality images, we know that it would be similarly limited in rendering time due to the volume rendering method used.
>
> We will address typos / spelling errors, and simplify the camera parameter notation to make the paper easier to follow.
>
> [1] Michael Oechsle, Songyou Peng, Andreas Geiger. “UNISURF: Unifying Neural Implicit Surfaces and Radiance Fields for Multi-View Reconstruction”. In ICCV, 2021.
>
> [2] Lior Yariv, Jiatao Gu, Yoni Kasten, Yaron Lipman. “Volume Rendering of Neural Implicit Surfaces”. arXiv preprint arXiv:2106.12052, 2021.
>
> [3] Peng Wang, Lingjie Liu, Yuan Liu, Christian Theobalt, Taku Komura, Wenping Wang. “NeuS: Learning Neural Implicit Surfaces by Volume Rendering for Multi-view Reconstruction”. arXiv preprint arXiv:2106.10689, 2021.

---

### Author Response · Authors · 2021-08-30
**Response Follow-Up**

Dear Reviewers,

Once again, we thank you for your time and effort in reviewing our paper. We hope that our response and additional experiments have cleared up many of your concerns. If there are any additional concerns or questions, please do not hesitate to let us know.

Thank you for your time and consideration,\
Paper2113 Authors

---

### Decision · Program_Chairs · 2021-09-27

**Decision:**

Accept (Poster)

**Comment:**

All four reviewers attest clarity of presentation, importance of the problem and solid engineering. There is a large spread in the judgements of the significance of the work explaining the different scores of 3,5,6,8. Overall the paper has been defended sufficiently well..